# Neuronal apoptosis drives remodeling states of microglia and shifts in survival pathway dependence

Sarah Rose Anderson[1], Jacqueline M Roberts[1], Nathaniel Ghena[1,2], Emmalyn A Irvin[1], Joon Schwakopf[1], Isabelle B Cooperstein[1], Alejandra Bosco[1], Monica L Vetter[1]*

[1]Department of Neurobiology, University of Utah, Salt Lake City, United States; [2]Interdepartmental Program in Neuroscience, University of Utah, Salt Lake City, United States

**Abstract:** Microglia serve critical remodeling roles that shape the developing nervous system, responding to the changing neural environment with phagocytosis or soluble factor secretion. Recent single-cell sequencing (scRNAseq) studies have revealed the context-dependent diversity in microglial properties and gene expression, but the cues promoting this diversity are not well defined. Here, we ask how interactions with apoptotic neurons shape microglial state, including lysosomal and lipid metabolism gene expression and dependence on Colony-stimulating factor 1 receptor (CSF1R) for survival. Using early postnatal mouse retina, a CNS region undergoing significant developmental remodeling, we performed scRNAseq on microglia from mice that are wild-type, lack neuronal apoptosis (Bax KO), or are treated with CSF1R inhibitor (PLX3397). We find that interactions with apoptotic neurons drive multiple microglial remodeling states, subsets of which are resistant to CSF1R inhibition. We find that TAM receptor Mer and complement receptor 3 are required for clearance of apoptotic neurons, but that Mer does not drive expression of remodeling genes. We show TAM receptor Axl is negligible for phagocytosis or remodeling gene expression but is consequential for microglial survival in the absence of CSF1R signaling. Thus, interactions with apoptotic neurons shift microglia toward distinct remodeling states and through Axl, alter microglial dependence on survival pathway, CSF1R.

*For correspondence: monica@neuro.utah.edu

Competing interest: The authors declare that no competing interests exist.

## Editor's evaluation

Your study highlights important and novel functions of neuronal apoptosis in shaping microglial states in the retina, as well as in the characterization of the underlying pathways. This work not only expands our understanding of core features of neural development but more broadly our comprehension of microglial biology across physiological and pathophysiological conditions. It will be of great interest to the field and the broad readership of *eLife*.

## Introduction

Microglia are parenchymal innate immune cells and vital remodelers of the central nervous system (CNS) (*Prinz et al., 2021*). They have a myriad of critical functions during development, including elimination of viable or dying cells (*Anderson and Vetter, 2019c*; *Anderson et al., 2019b*). Similar to other macrophages, microglia are extremely attuned to changes in their neural niche (*Lavin et al., 2014*; *Gosselin et al., 2017*; *Bennett et al., 2018*) and rapidly respond by phagocytosis or secreting soluble factors. Recent single-cell RNA sequencing (scRNAseq) has demonstrated that microglia are

particularly diverse during development (*Hammond et al., 2019*; *Li et al., 2019a*). However, we still lack an understanding of the drivers of this heterogeneity, including the impact of environmental cues or the act of phagocytosis itself. Therefore, connecting microglial states to specific developmental events and determining the pathways involved remains a central challenge.

One developmental process fundamental for nearly every organ in the body is apoptotic death of excess or dysfunctional cells (*Jacobson et al., 1997*). In the CNS, death of neurons and glia is an essential and well-documented feature of development (*Oppenheim, 1991*). Efferocytosis, or the phagocytosis of apoptotic cells, is a rapid and carefully orchestrated process designed to minimize damage to surrounding cells (*Morioka et al., 2019*; *Boada-Romero et al., 2020*). Largely mediated by microglia, efferocytosis is important for maintaining CNS homeostasis not only in development but in aging and disease as well (*Galloway et al., 2019*; *VanRyzin, 2021*). The developing mouse retina is an excellent CNS model system to link remodeling events such as efferocytosis and microglial state and function (*Li et al., 2019b*). During the postnatal period in the retina, as circuits are being established, waves of neuronal death occur (*Braunger et al., 2014*). For example, retinal ganglion cells (RGCs), the projection neurons connecting retina and brain, undergo a well-characterized culling during the first postnatal week which is dependent on pro-apoptotic factor, Bcl-2-associated X protein (Bax) (*Péquignot et al., 2003*). An estimated ~50% of RGCs will undergo apoptosis and need to be cleared (*Farah and Easter, 2005*; *Farah, 2006*). We previously found that neuronal death significantly influences microglia properties in postnatal retina, resulting in a majority of microglia expressing a distinct gene signature and showing altered dependence on CSF1R signaling for survival (*Anderson et al., 2019a*). Relative to adult microglia, homeostatic genes were reduced, while genes associated with phagocytosis and lipid metabolism were increased, similar to aging and disease-associated microglia (DAM) (*Holtman et al., 2015*; *Keren-Shaul et al., 2017*; *Krasemann et al., 2017*), and to developmental microglia residing in postnatal white matter tracts (PAM) (*Li et al., 2019a*) or (ATM) (*Hammond et al., 2019*) or CD11c microglia (*Benmamar-Badel et al., 2020*).

Consistent with either apoptotic cell recognition or phagocytosis playing a central role in driving this gene signature, microglia in disease states increase select DAM genes in response to apoptotic cells (*Krasemann et al., 2017*), and ATM/PAM populations in the white matter tract engulf oligodendrocyte precursor cells (*Li et al., 2019a*; *Nemes-Baran et al., 2020*). In disease, the TREM2 receptor is partially required for acquisition of the DAM signature (*Keren-Shaul et al., 2017*; *Krasemann et al., 2017*), but not for the developmental signature in retina (*Anderson et al., 2019a*) or brain (*Li et al., 2019a*). Thus, it remains unclear to what extent the process of phagocytosis is altering properties of developmental microglia or whether distinct recognition pathways are involved. Microglia, like other macrophages, can recognize and engulf apoptotic (or viable) cells via a variety of ligand-receptor systems including C1q/C3 to complement receptor 3 (CR3) and exposed phosphatidylserine (PtdSer) to TAM family of tyrosine kinase receptors Mer (gene Mertk) and Axl (*Lemke, 2019*), but how these pathways mediate changes in microglia properties and responses is an active area of research.

Here, we ask how neuronal apoptosis drives key properties of developmental microglia such as CSF1R dependence and expression of DAM/ATM/PAM-related genes, and test the role of phagocytosis and specific recognition pathways. Through scRNAseq on postnatal retinal microglia, we find considerable transcriptional heterogeneity, with multiple distinct microglial populations expressing DAM/ATM/PAM-related, lysosomal, and lipid metabolism genes. We further show, by analyzing microglia from Bax knockout (KO) retinas, that exposure to dying neurons drives several of these states and that most are more resistant to CSF1R inhibition. We find that CR3 and Mer are required for clearance of dying RGCs, but not for altering dependence on CSF1R signaling for survival. Conversely, Axl is not required for clearance of dying RGCs, but for augmenting microglial survival in the absence of CSF1R signaling. Loss of Mer did not have a widespread impact on expression of microglial remodeling genes, and neither did loss of Axl or Mertk/Axl double KO (dKO), suggesting that TAM receptor-mediated signaling or reduced clearance are not sufficient. Thus, we find that interactions with apoptotic neurons drives developmental microglial diversity, and that distinct recognition receptors mediate phagocytosis of dying cells versus specific microglial properties, including microglial survival.

# Results

## Multiple microglial states coexist in postnatal retina

To better understand the influence of environmental cues on microglial states, we first sought to understand the extent of microglial heterogeneity in the postnatal retina and then explicitly determine states that were driven by neuronal apoptosis. Second, we previously linked altered CSF1R dependence to a DAM/ATM/PAM-related signature and neuronal apoptosis (*Anderson et al., 2019a*), so we wanted to identify and characterize microglial states that remained following CSF1R inhibition (PLX3397, PLX). Therefore, we performed scRNAseq on sorted retinal microglia from four groups at postnatal day 6 and 7 (P6/P7): Bax WT, Bax KO, CX3CR1-GFP/ + given vehicle (daily for 3 days), and CX3CR1-GFP/ + pups dosed with PLX3397 (daily for 3 days) (*Figure 1*). Microglia from Bax WT (26 retinas) and littermate KOs (26 retinas) (CD45$^+$ CD11b$^+$/CX3CR1-GFP$^+$ CCR2$^-$) as well as microglia from PLX (24 retinas) and vehicle controls (22 retinas) (CD45$^+$ CX3CR1-GFP$^+$ Ly6C$^-$) were sorted (*Figure 1—figure supplement 1A-D*) and sequenced using the 10 X Genomics platform. Following sequencing and manual filtering (*Figure 1—figure supplement 2*), we used in silico Bax genotyping to sort out and reassign a subset of cells from animals incorrectly partitioned to the Bax WT and KO samples (*Figure 1—figure supplement 3*). Unsupervised clustering was performed on the 15,084 cells from the four groups (4902 - Bax WT; 4224 - Bax KO; 4198 - Vehicle; 1760 - PLX3397) (*Figure 1—figure supplement 4A*). Of the 15,084 cells, 1417 in satellite clusters were deemed non-microglia cells based on expression of established markers and were excluded from further analysis (*Figure 1—figure supplement 4B-E*). Differences in cell surface markers used to sort Bax WT/KO (CD45$^+$CD11b$^+$CX3CR1$^+$CCR2$^-$) compared to PLX/Vehicle (CD45$^+$CX3CR1$^+$Ly6C$^-$) (*Figure 1—figure supplement 1*) led to variance in the proportion of macrophage/monocyte-like cells (*Figure 1—figure supplement 4F*), but this had no major impact on microglial cell number (Gray dots, *Figure 1—figure supplement 4C, F*).

Unsupervised re-clustering of 13,667 remaining cells yielded 11 microglia clusters (*Figure 1*). Cells from controls (Bax WT and Vehicle) were represented in every cluster (*Figure 1*), indicating that microglia in an individual developing CNS region during a specific time window are strikingly diverse. Further, we found that Bax KO and PLX-treated microglia had dramatic and opposing shifts in their distribution compared to controls (*Figure 1*). To determine the characteristics of the 11 clusters, we examined the top genes for each (*Figure 1*, *Supplementary file 1*). While virtually every cell expressed *Apoe* and *Ctsd*, both genes were significantly enriched in cluster 0, which had high expression of other DAM (*Keren-Shaul et al., 2017*), PAM (*Li et al., 2019a*), and ATM (*Hammond et al., 2019*) genes, such as *Lyz2* and *Cd9*, but also intermediate levels of homeostatic gene expression such as *P2ry12* and *Tmem119*. Thus, we called this cluster Apoe-enriched. Cluster 1 had highest expression of *Tmem119* and *P2ry12* as well as other homeostatic genes, including *Itgam, Siglech, Tgfbr1, P2ry13, Selplg, and Adgrg1* and so we named it Homeostatic. Cluster 2 was enriched for chemokines including *Ccl4* and *Ccl3*, two genes previously found in injury-responsive microglia in demyelinated lesions (*Hammond et al., 2019*) and in PAM of developing WM tracts (*Li et al., 2019a*), and had intermediate/high expression of homeostatic genes (e.g. *Tmem119*; *Figure 1*) and so we termed it Hom/chemokine cluster. *Ccl3* and *Ccl4* are among a set of genes that can be induced by dissociation (*Marsh et al., 2020*) so we confirmed that retinal microglia express *Ccl3* and *Ccl4* in vivo by in situ hybridization (*Figure 1—figure supplement 5* and data not shown). Cluster 3 was marked by *Npl*, found in lipid-droplet microglia (*Marschallinger et al., 2020*), and *Apoc1*, a lipoprotein high in multiple sclerosis-associated human microglia (*Masuda et al., 2019*). Since this cluster was substantially increased in PLX-treated retinas, we named this cluster PLX-enriched. Chemokines *Cxcl2* and *Cxcl10* and pro-inflammatory cytokines, such as *Il1b* and *Tnf*, were high in Cluster 4, dubbed the Chemokine/cytokine cluster. Cluster 5 most resembled ATM (*Hammond et al., 2019*) (69/193 genes shared) and PAM (*Li et al., 2019a*) (79/193 genes shared) with specific expression of *Spp1* and high expression of *Fabp5, Ctsl, Lpl, Igf1, and Csf1* and thus we termed it ATM/PAM-like. Cluster 6 was named Antioxidant-responsive cluster due to high expression of antioxidant responsive genes, *Hmox1* and *Gclm* (*Rojo et al., 2014*). Clusters 7, 8, and 9 represented Cycling microglia and expressed *Mki67, Top2a*, and *Mcm* genes. Cluster 10, the smallest, contained 71 cells with high and very specific expression of interferon response genes such as *Ccl5, Ifit3*, and *Ifitm3* (*Dorman et al., 2022*) and was thus termed Interferon-responsive cluster. Overall, these proportions were consistent with our previous quantification of select homeostatic and DAM-related genes by in situ hybridization (*Anderson et al., 2019a*). Thus, these data suggest that within the developing retina, several

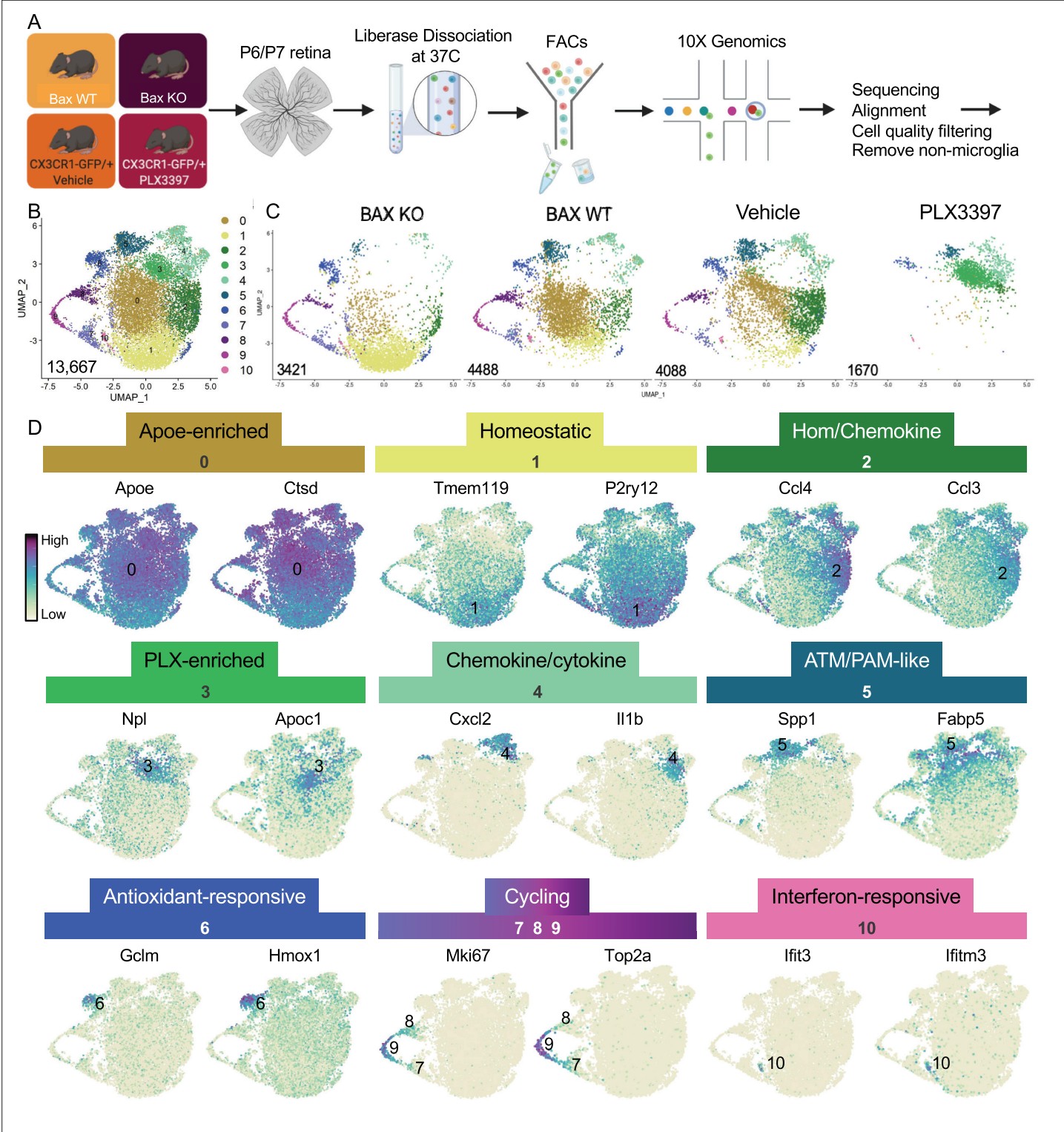

**Figure 1.** Multiple microglial states coexist in postnatal retina. (A) Workflow for collection, dissociation, sorting, sequencing, and filtering of individual microglia from four different groups. 13 P6/P7 animals from 6 litters (26 retinas) pooled for Bax WT and KO samples, 12 P6/P7 animals from 2 litters (24 retinas) for PLX, and 11 animals from 2 litters (22 retinas) for Vehicle. (B) UMAP plot of 13,667 microglia cells from all 4 samples distributed into 11 clusters by unsupervised clustering. Blue cells to the right of cluster 1 are members of cluster 6. (C) UMAP plots illustrating the distribution of cells from each condition. Number of cells per condition labeled in lower left. (D) UMAP plots of two genes enriched in each cluster. Color scale is based on relative gene expression: dark purple = highest, light yellow = lowest.

*Figure 1 continued on next page*

Figure 1 continued

The online version of this article includes the following figure supplement(s) for figure 1:

**Figure supplement 1.** Gating strategy for FAC-sorting retinal microglia for single-cell sequencing.

**Figure supplement 2.** Selection of high-quality cells.

**Figure supplement 3.** In Silico *Bax* genotyping.

**Figure supplement 4.** Identification of non-microglia populations.

**Figure supplement 5.** Postnatal retinal microglia express *Ccl3* in vivo.

microglial states coexist including homeostatic and various subsets of DAM/PAM/ATM-related and chemokine-expressing microglia.

## Postnatal retinal microglia encompass a spectrum of states from homeostatic to remodeling

We noted a continuum of expression of the homeostatic genes *Tmem119* and *P2ry12* across the clusters that appeared to be inverse to broadly expressed DAM/PAM/ATM genes such as *Apoe* and *Ctsd* (*Figure 1*). We more closely examined the expression of homeostatic genes, as well as genes associated with lysosomal function, lipid metabolism and transport, and other DAM/PAM/ATM-related genes (see methods for lists). As for *Tmem119* and *P2ry12*, we found a gradient of expression of homeostatic genes including *Siglech, Tgfbr1*, and *Selplg* which were highest in Homeostatic cluster (1), high/intermediate in Hom/chemokine (2), intermediate in Apoe-enriched (0), Chemokine/cytokine cluster (4), ATM/PAM-like (5), Antioxidant-responsive (6), and Interferon-responsive (10), with lowest expression in PLX-enriched cluster (3), suggesting a spectrum from more to less homeostatic (*Figure 2C and C'*).

To determine whether this related to shifting phagocytic function of microglia, we examined the expression of lysosomal genes including *Lyz2, Galns* and *Cst7* and saw roughly the opposite pattern. These genes were either low, absent, or sparse in Homeostatic cluster (1) and Hom/chemokine (2) and generally more highly or broadly expressed in Apoe-enriched (0), PLX-enriched (3), Chemokine/cytokine cluster (4), ATM/PAM-like (5), Antioxidant-responsive (6), and Interferon-responsive (10) (*Figure 2D and D'*). Notably, *Lyz2* showed highest levels in PLX-enriched (3) the least homeostatic cluster. Comparing our clusters to a gene ontology (GO) list of 552 Lysosomal genes (GO:0005764) (*Bult et al., 2019*), we found cluster 3 had the greatest enrichment (35/552), followed by clusters 5, 10, 6, 4, and 0, with homeostatic clusters (1 and 2) expressing the lowest number of genes. Additionally, genes involved in lipid metabolism and cholesterol transport, important processes following the engulfment of apoptotic cells (*Doran et al., 2020*), followed the same trend (*Figure 2E and E'*). Cluster 3 again expressed the highest number (36/1428) of Lipid Metabolic process genes (GO:0006629) (*Bult et al., 2019*), followed by 5, 6, 4, and 10 while homeostatic clusters (1 and 2) expressed the lowest number. Thus, microglia in clusters (0,3,4,5,6,10) are equipped to be actively breaking down phagocytosed material compared to more homeostatic clusters (1,2). Therefore, we hypothesize that clusters 0,3,4,5,6, and 10 are involved in active remodeling activities and refer to them as remodeling clusters (*Figure 2B*). We confirmed genes involved in lysosomal and lipid metabolism function were enriched in remodeling clusters by differential expression analysis on clusters (1,2) compared to (0,3,4,5,6,10) (*Supplementary file 2*). We next examined the expression of additional DAM/ATM/PAM-related genes, including *Igf1*, and recognition receptors *Itgax* and *Axl*. We noted similar low expression in Homeostatic cluster (1) and Hom/chemokine (2) and higher expression in the remodeling clusters (0,3,4,5,6,10) (*Figure 2F and F'*). Interestingly, DAM/PAM/ATM-related genes had varying cluster specificity. Genes such as *Apoe* were expressed in nearly every cell (*Figure 1*), whereas *Igf1* and *Spp1* were more restricted to specific clusters, suggesting that regulation of these genes is complex (*Figure 2F*). Together, these results suggest that phagolysosomal function defines the spectrum of microglial states in the postnatal retina.

## Neuronal apoptosis drives multiple remodeling states

We previously showed that neuronal death has a major influence on expression of DAM-related genes in postnatal retinal microglia (*Anderson et al., 2019a*) and wondered whether recognition or clearance of dying neurons was a key factor in driving diverse microglial remodeling states. We examined

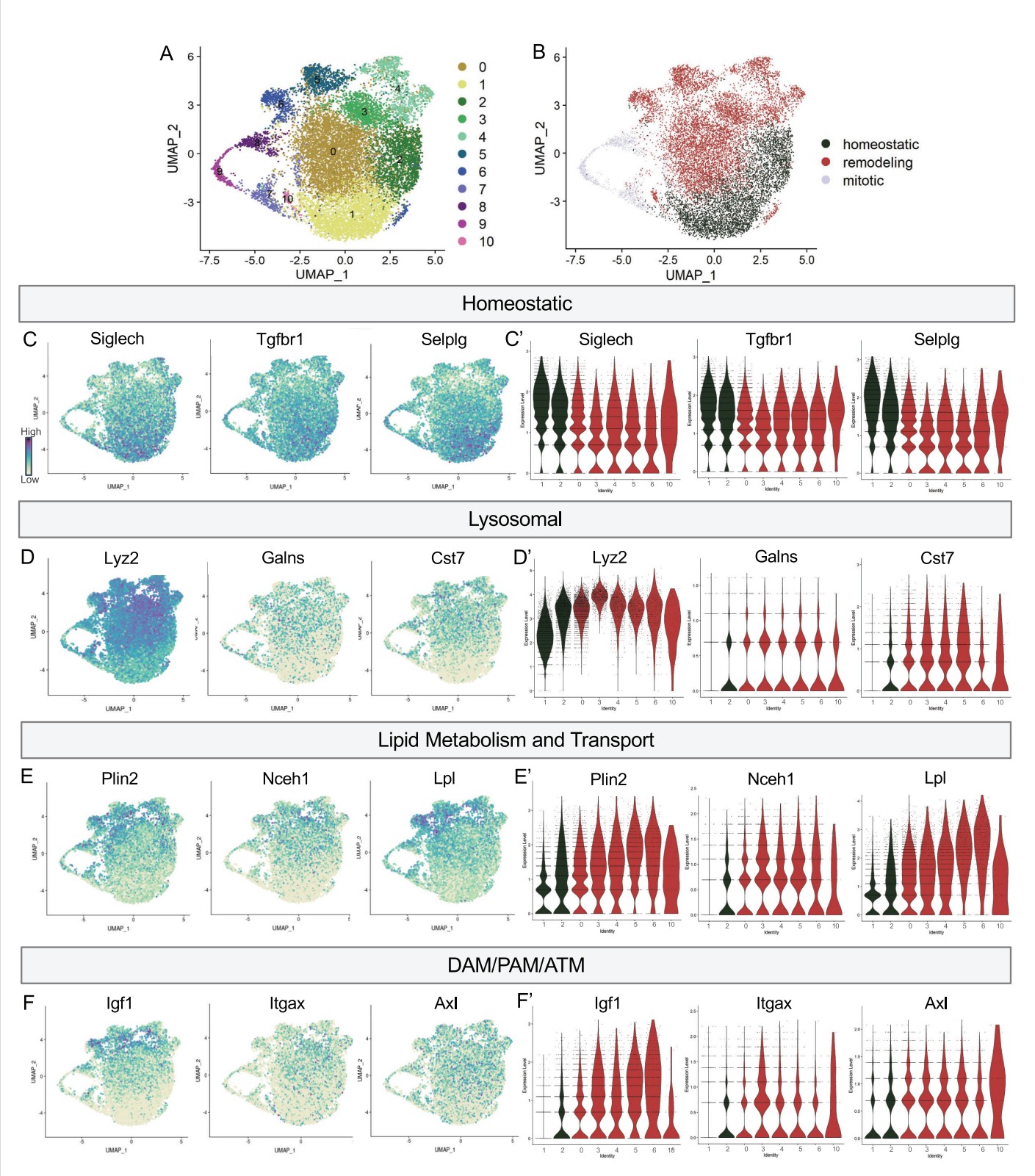

**Figure 2.** Postnatal microglia encompass a spectrum of states from homeostatic to remodeling. (**A**) UMAP plot of 11 clusters from all sequenced microglia. (**B**) UMAP plot identifying clusters assigned homeostatic, remodeling, and mitotic. (**C**) UMAP and (**C′**) violin plots of microglial homeostatic genes. (**D**) UMAP and (**D′**) violin plots of select genes important for lysosomal function. (**E**) UMAP and (**E′**) violin plots of genes associated with lipid metabolism. (**F**) UMAP and (**F′**) violin plots of DAM/PAM/ATM genes. Color scale for UMAP plots is based on relative gene expression: dark purple = highest, light yellow = lowest.

microglia from Bax KO retinas in which apoptotic death programs in RGCs and other neurons are selectively lost (*Fricker et al., 2018*) and saw a dramatic shift in cluster distribution with the loss of Bax compared to littermate controls (*Figure 3A*). We found a fivefold expansion in the Homeostatic cluster (1) in Bax KO compared to WT (*Figure 3A, B and C*). This was concurrent with decreases in remodeling clusters Apoe-enriched (0), Chemokine/cytokine expressing (4), and ATM/PAM-like (5), and the minor PLX-enriched cluster (3), which were 4.7-fold, 3.46-fold, 5.86-fold, and fivefold more abundant in WT, respectively (*Figure 3A and B*). Clusters that remained largely unaltered included Hom/chemokine (2), Antioxidant-responsive (6), Cycling (7–9), and Interferon-responsive (10), suggesting they were not driven by apoptotic cell interactions or clearance (*Figure 3A and B*). Therefore, we conclude that the spectrum of homeostatic to more remodeling clusters (0,3,4, and 5) in the postnatal retina is driven predominately by neuronal apoptosis.

To further examine microglial genes dependent on neuronal apoptosis, we compared all cells from each genotype, ignoring cluster membership, and identified genes that were specific to each sample (*Figure 3D*, *Supplementary file 3*). Consistent with a shift to a more homeostatic state, *Tmem119, P2ry12,* and *Siglech* were among the 133 upregulated genes in microglia from Bax KO retina. The 183 downregulated genes included lysosomal and lipid metabolism genes such as *Apoe, Lyz2, Cst7, Ctsb, Lpl* and *Cd68*, DAM/PAM/ATM-related genes such as *Igf1* and *Spp1,* and recognition receptors *Itgax* and *Axl* (*Figure 3C and D*). We next performed KEGG analysis using GeneCodis 4 (*García-Moreno et al., 2021*) on downregulated genes in Bax KO to identify major pathways driven by interactions with dying neurons (*Figure 3E*). Consistent with the idea that microglial phagocytosis was a key factor in these gene expression changes, we found metabolic and lysosomal pathways significantly reduced. To determine whether loss of Bax altered microglia density or distribution, we used C1q as a marker for microglia, which we validated by both scRNAseq (*Figure 1—figure supplement 4C*) and immunostaining (*Figure 3—figure supplement 1A*,B). We found that microglia remained uniformly spaced but that density was reduced by nearly half in Bax KO retinas (*Figure 3—figure supplement 1C*,D), consistent with our prior flow cytometry analysis (*Anderson et al., 2019a*). Altogether, this suggests that neuronal apoptosis regulates important properties of microglia in the postnatal retina, including overall density as well as the emergence of multiple remodeling states.

## Subsets of remodeling states survive CSF1R inhibition, while homeostatic microglia are more vulnerable

A striking property of microglia in postnatal retina is that ~60% survive inhibition of CSF1R signaling, while only very small subsets show this property in postnatal brain (*Anderson et al., 2019a*) or adult brain (*Zhan et al., 2020*). We previously found that surviving microglia had reduced homeostatic gene expression and increased DAM/PAM/ATM gene expression, and this required neuronal apoptosis (*Anderson et al., 2019a*). Since we found that the less homeostatic remodeling clusters (0,3,4,5) were driven by neuronal apoptosis, we investigated whether these microglial states would be more resistant to CSF1R inhibition. We examined scRNAseq of Vehicle and PLX-treated microglia (dosed for 3 days) and found a large shift in the distribution of cells across clusters with PLX treatment (*Figure 4A and B*). Several remodeling clusters were either enriched or maintained. The least homeostatic cluster, PLX-enriched (3), which was a very small proportion of cells in both Vehicle and Bax WT (*Figure 3A and B*), represented nearly 60% of remaining microglia following PLX3397 (increased sixtyfold) and thus seemed to arise, in part, by treatment (*Figure 4A and B*). This could potentially be due to the further reduction of homeostatic properties of surviving microglia with CSF1R blockade (*Kempthorne et al., 2020*). Remodeling cluster Chemokine/cytokine (4) was increased twofold with PLX-treatment and ATM/PAM-like (5), Antioxidant-responsive (6), and Interferon-responsive (10) were slightly reduced or unchanged (0.755-fold, 0.654-fold, 0.979-fold, respectively), suggesting they are more resistant than other subsets (*Figure 4A and B*). Conversely, we found clusters that had high or intermediate expression of homeostatic genes, Apoe-enriched (0), Homeostatic (1), Hom/chemokine (2), and Cycling (7,8,9) were largely absent following PLX treatment, suggesting these clusters are more dependent on CSF1R for survival or that they had shifted to a different state (*Figure 4A and B*). We noted the overlap between clusters dependent on neuronal apoptosis (0,3,4, and 5) and resistant to loss of CSF1R signaling (3,4,5,6, and 10), illustrating the link between the two and arguing that certain microglial states driven by neuronal apoptosis may confer resistance to CSF1R inhibition. Furthermore, we found that downregulation of *Csf1r* alone does not alter dependence as more

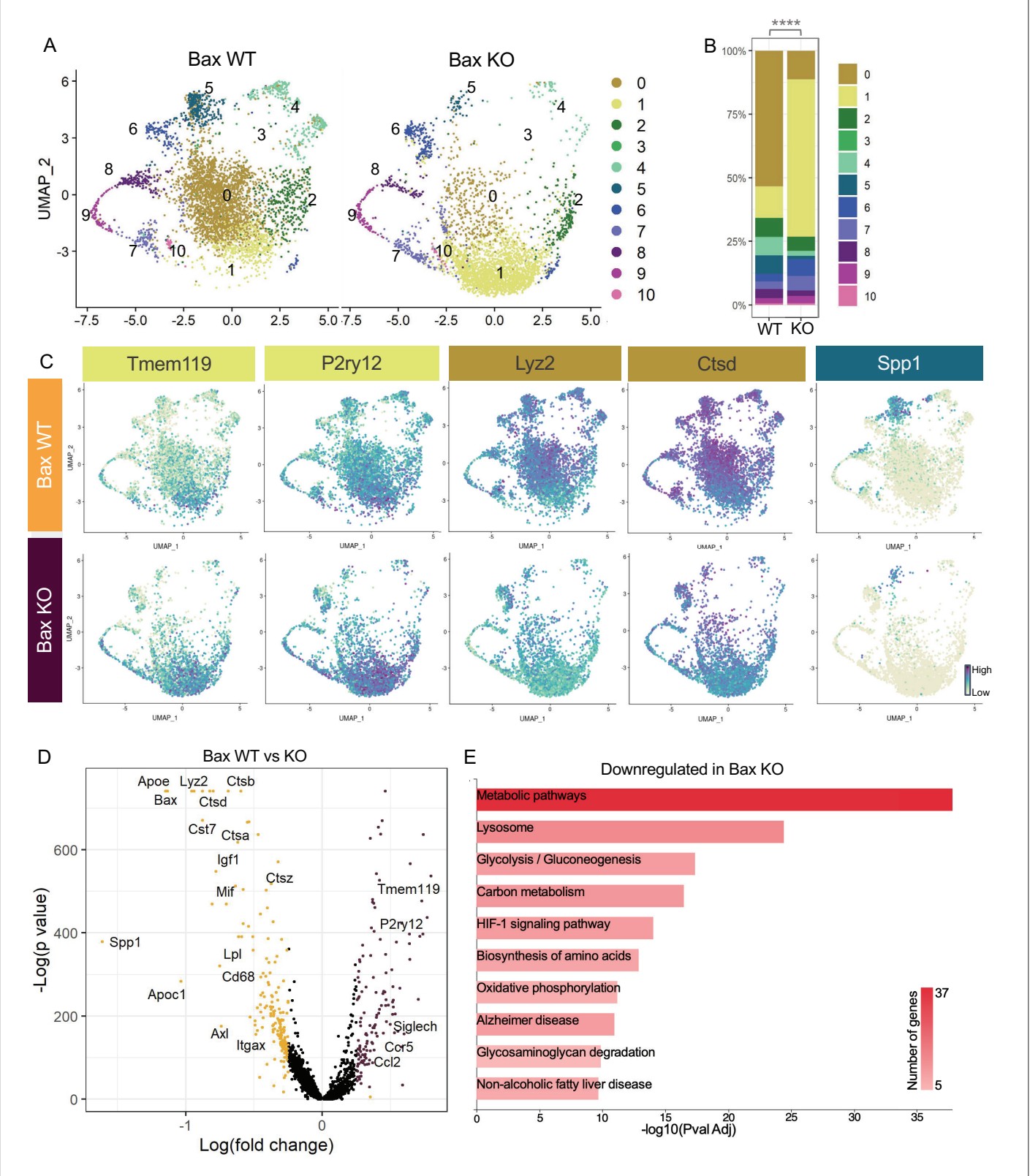

**Figure 3.** Neuronal apoptosis drives multiple microglial remodeling states. (**A**) UMAP plot of microglia cells from Bax WT (left) and Bax KO (right) samples distributed into 11 clusters. (**B**) Bar graph of the proportion of cells in each cluster for each sample. Chi-square test comparing cluster distribution ****p < 0.0001. (**C**) UMAP plots showing expression of representative genes from selected clusters. (**D**) Volcano plot showing differential gene expression of all Bax KO cells compared to Bax WT cells. Each gene is plotted according to the significance (-Log(p value)) and magnitude

*Figure 3 continued on next page*

Figure 3 continued

(Log(fold change)) of the difference such that those genes enriched in Bax KO are colored purple, and those down-regulated in Bax KO are yellow. Differentially expressed genes are defined by p-value ≤ 0.05 and absolute value of Log(fold change) > 0.25. (**E**) KEGG Pathway analysis of 183 downregulated genes in Bax KO compared to Bax WT using GeneCodis 4.

The online version of this article includes the following figure supplement(s) for figure 3:

**Figure supplement 1.** Bax KO retinas have reduced microglia density.

resistant clusters (3,4,5,6, and 10) had slightly reduced expression of *Csf1r* compared to susceptible clusters (0,1, and 2) but comparable to other susceptible clusters (7,8,9) (*Figure 4D*), consistent with our previous findings (*Anderson et al., 2019a*).

To further probe genes and pathways enriched following CSF1R inhibition, we performed a pairwise comparison on all Vehicle and PLX-treated cells, regardless of cluster identity. We found that DAM/ATM/PAM-related genes *Apoe, Itgax,* and *Fabp5,* lysosomal genes *Lyz2* and *Ctss,* and lipid metabolism genes *Nceh1, Soat1, Abca1,* and *Apoc1* were among the 254 upregulated genes in PLX-treated cells (*Figure 4C, E, Supplementary file 4*). Consistent with a loss of more homeostatic microglia, homeostatic genes including *Tmem119, P2ry12,* and *Siglech* were significantly reduced (181 genes downregulated) (*Figure 4C, E*). To uncover pathways associated with resistance to CSF1R inhibition, we performed KEGG analysis (*García-Moreno et al., 2021*) on all upregulated genes in PLX cells and found that translation, metabolic pathways, and phagosome processes were enriched (*Figure 4F*). Altogether, we conclude that a change in CSF1R dependence is linked to increased phagocytosis, lysosomal function, and altered microglia metabolic states, and that homeostatic microglia are more dependent on CSF1R signaling for survival.

## Mer and CR3 are required for apoptotic RGC clearance

To understand the role of apoptotic cell phagocytosis in promoting distinct microglial properties, we sought to test whether recognition receptors were important both for driving microglial remodeling states as well as changes in CSF1R dependence. We first wanted to confirm that microglia were important for clearance of dying neurons within the retina and identify the recognition receptors required, focusing on apoptotic RGCs. We analyzed genetic KO of receptors previously implicated in the finding, recognition, or phagocytosis of apoptotic cells: find-me pathway, fractalkine receptor CX3CR1 (*Wolf et al., 2013*), integrin receptor complement receptor 3 (integrin $\alpha_M\beta_2$, CD11b, CR3), and TAM receptors Mer and Axl (*Figure 5A*; *Fourgeaud et al., 2016*; *Lemke, 2019*). We used wildtype animals from the CX3CR1 (CX3CR1-WT) and Mer (Mertk-WT) background as well as CX3CR1-GFP/+ (*Jung et al., 2000*) for controls. First, we looked at whether loss of any of these receptors resulted in changes to microglial density. By wholemount immunostaining, we found slight variations at P5 but none that reached significance compared to CX3CR1-GFP/+ (*Figure 5—figure supplement 1A, B*). Next, we analyzed the density of total apoptotic bodies by cleaved caspase 3 (CC3) in the ganglion cell layer after the peak of RGC death to measure any buildup (*Figure 5—figure supplement 1C, D* and *Figure 5—figure supplement 2A*). We found that loss of CX3CR1, CR3, Mer, Axl, and both Mer/Axl resulted in increased apoptotic body density compared to WT controls suggesting these pathways were important for microglial phagocytosis of dying cells within the retina. To ask what pathways were important for clearing dying RGCs specifically, we looked at the density of CC3+RBPMS+ double-positive cells at P5 in all KOs (*Figure 5B, C and D*, *Figure 5—figure supplement 2B*). We found the CR3 KO, Mertk KO, and Mertk/Axl dKO all had increased density of dying RGCs compared to controls (*Figure 5C and D*) with no change in overall RGC density or retinal blood vessel development (*Figure 5—figure supplement 1E*,F,G,H). CX3CR1 and Axl were dispensable for clearance of dying RGCs, and Mertk/Axl dKOs did not appear to have a further clearance deficit above Mertk KO alone (*Figure 5C, D*). Thus, both CR3 and Mer receptors, which are broadly expressed in microglia, are important for the timely clearance of RGCs undergoing developmental cell death, consistent with prior studies implicating these pathways in efferocytosis in the CNS (*Lemke, 2019*).

## Axl promotes microglial survival in the absence of CSF1R signaling

Having identified receptors important for RGC clearance (CR3 and Mer), we were next able to ask whether these pathways were also important for driving microglial survival following CSF1R inhibition.

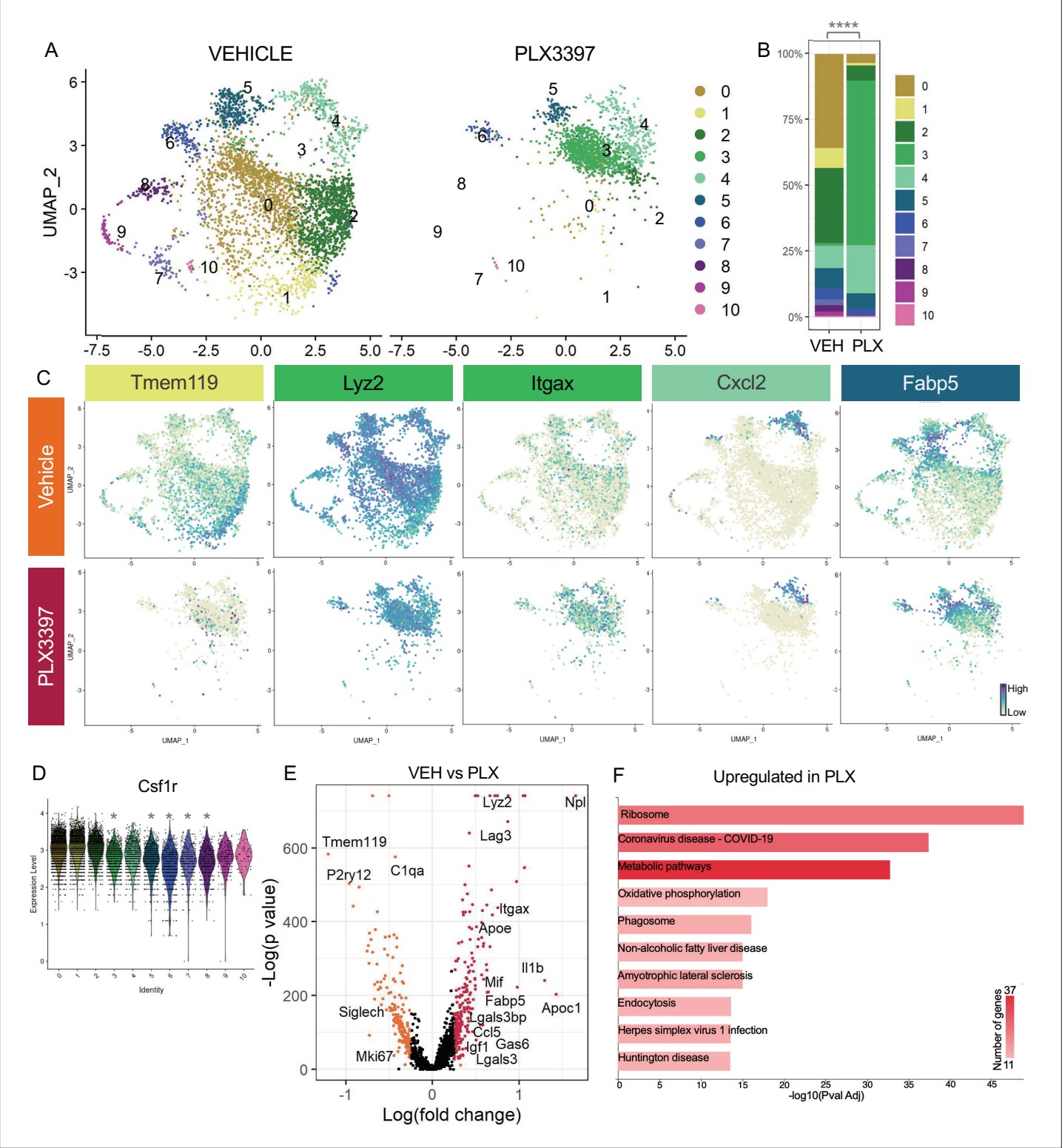

**Figure 4.** Subsets of remodeling states survive CSF1R inhibition, while homeostatic microglia are more vulnerable. (**A**) UMAP plot of microglia cells from Vehicle (left) and PLX3397 (right) samples distributed into 11 clusters. (**B**) Proportion of cells from each sample across 11 clusters. Chi-square test comparing cluster distribution ****p < 0.0001. (**C**) UMAP plots showing expression of representative genes from selected clusters. (**D**) Violin plot of *Csf1r* expression across clusters. Asterisks mark those clusters with reduced Csf1r expression when compared to all other cells. *padj <0.0001. (**E**) Volcano plot showing differential gene expression of all PLX cells compared to Vehicle cells. Each gene is plotted according to the significance (-Log(p value)) and magnitude (Log(fold change)) of the difference such that those genes enriched in PLX are colored red, and those downregulated in PLX are

*Figure 4 continued on next page*

Figure 4 continued

orange. Differentially expressed genes are defined by p-value ≤ 0.05 and absolute value of Log(fold change) > 0.25. (**F**) KEGG Pathway analysis of 254 upregulated genes in PLX compared to Vehicle using GeneCodis 4.

We administered PLX3397 as previously published (*Anderson et al., 2019a*; *Figure 6A*) and, surprisingly, found that loss of CR3, CX3CR1, or Mer did not significantly reduce the proportion of surviving microglia compared to controls (*Figure 6B, C*). However, loss of Axl allowed for greater depletion, matching levels achieved with loss of developmental apoptosis (Bax KO) that we previously reported (*Anderson et al., 2019a*; *Figure 6B, C*). Mertk/Axl dKOs did not have an additive effect, suggesting

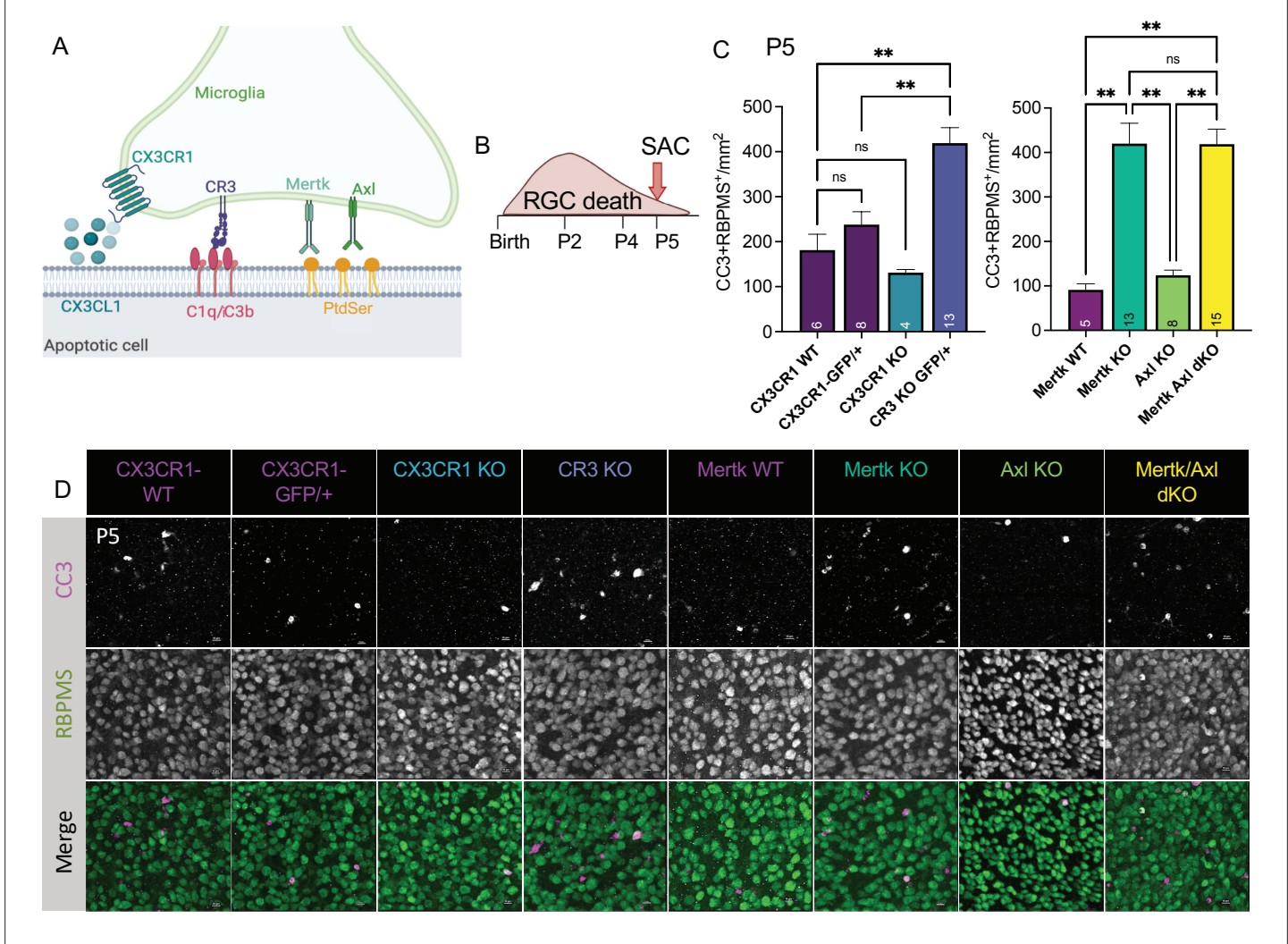

**Figure 5.** Mer and complement receptor 3 (CR3) are required for apoptotic retinal ganglion cell (RGC) clearance. (**A**) Cartoon of candidate pathways. (**B**) Schematic for collection at P5 after the bulk of RGC developmental death. (**C**) Quantification of the average of central and peripheral 0.4 mm² RGC death (*Figure 5—figure supplement 2B*) in dorsal leaf of all genotypes. (left) (n = 6 CX3CR1 WT, n = 8 CX3CR1-GFP/+, n = 4 CX3CR1 KO, n = 13 CR3 KO, CX3CR1-GFP/+; ± SEM) ≥ 2 litters collected for each genotype. Welch's ANOVA test W(3,12.58) = 23.75, p < 0.0001 and Dunnett's T3 multiple comparisons tests. (right) (n = 5 Mertk WT, n = 13 Mertk KO, n = 8 Axl KO, n = 15 Mertk Axl dKO; ± SEM) ≥ 2 litters collected for each genotype. Kruskal-Wallis test statistic = 25.97 p < 0.0001 and Dunn's multiple comparisons tests. Not all comparisons shown on graphs but can be found in *Supplementary file 7*. (**D**) Max projected confocal images of dying RGCs (CC3⁺RBPMS⁺) in KOs in the dorsal mid-periphery in the ganglion cell layer. Apoptotic bodies, CC3 (magenta); RGCs, RBPMS (green). Scale bars 10 µm.

The online version of this article includes the following figure supplement(s) for figure 5:

**Figure supplement 1.** Loss of candidate receptors does not alter microglial density, RGC density, or blood vessel development.

**Figure supplement 2.** Analysis of CC3 and CC3/RBPMS density.

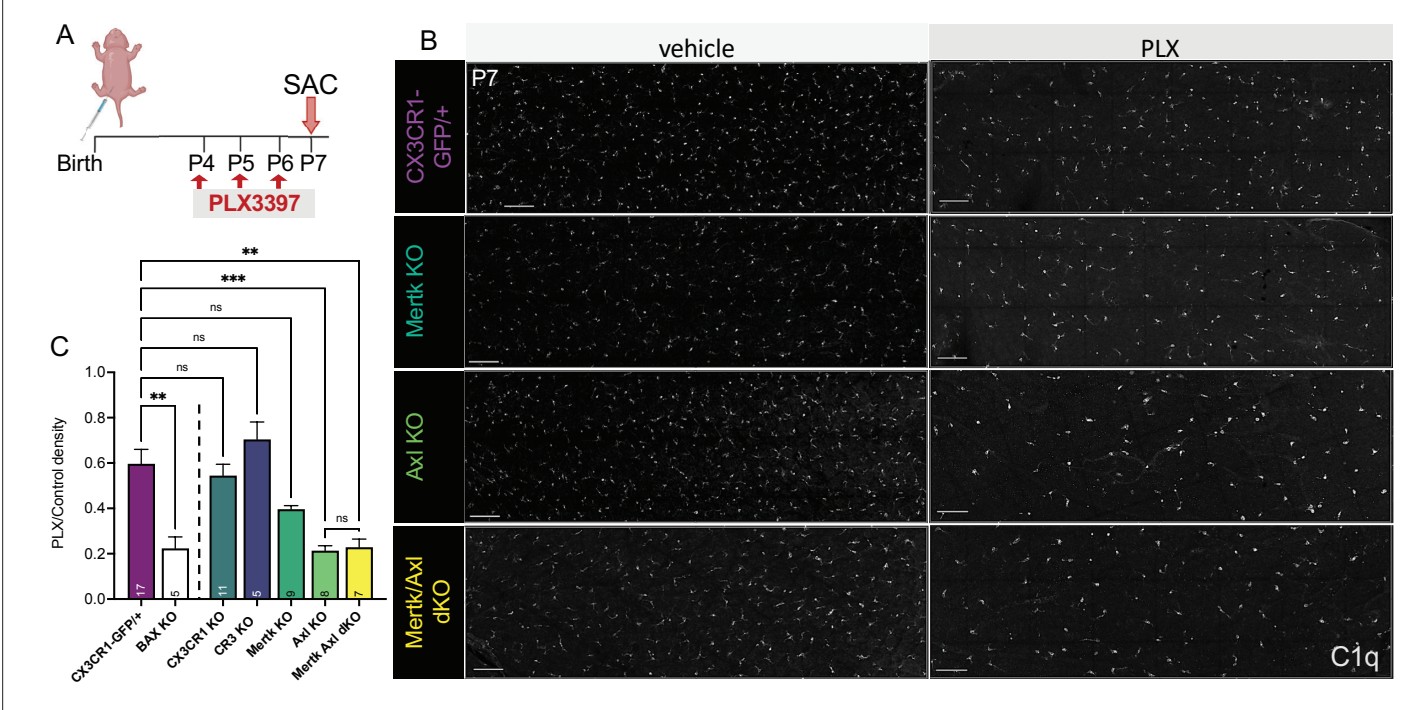

**Figure 6.** Axl signaling promotes microglial survival in the absence of CSF1R signaling. (**A**) Dosing regimen of PLX3397 to various genotypes. (**B**) Confocal images of microglia in the NFL/GCL in all genotypes from central to mid-periphery of the dorsal leaf. C1q (mono). Scale bars 100 μm. (**C**) Ratio of density of CD45+CX3CR1-gfp+ or CD45+CD11b+ (microglia/singlets) in PLX treated retinas over genotype-matched controls by flow cytometry. (n = 17 CX3CR1-GFP/+, n = 5 Bax KO, n = 11 CX3CR1 KO, n = 5 CR3 KO CX3CR1-GFP/+, n = 9 Mertk KO, n = 8 Axl KO, n = 7 Mertk Axl dKO; ± SEM) ≥ 2 litters collected for each genotype. Line demarcates data from CX3CR1-GFP/ + and Bax KO previously published in *Anderson et al., 2019a*. Welch's ANOVA W(6,18.55) = 16.53, p < 0.0001 and Dunnett's T3 multiple comparisons test. Not all comparisons shown but can be found in *Supplementary file 7*.

Axl signaling alone was important for changes in microglial survival in the absence of CSF1R signaling (*Figure 6B, C*). Altogether, we find that the receptors important for effective clearance (Mer and CR3) do not alter microglial dependence on CSF1R, but that TAM receptor Axl, which is induced by neuronal apoptosis, augments microglial survival in the absence of CSF1R signaling in retina.

## Mer and Axl are not required for expression of lysosomal, lipid metabolism, or remodeling genes

Since Mer was important for phagocytosis while Axl mediated survival in the absence of CSF1R signaling, we wanted to delineate microglial gene expression changes that were driven by Mer versus Axl. We performed bulk RNA-seq on sorted microglia from P4 Mertk and Axl KOs and compared them to WT controls by DESeq2 (*Figure 7A*). Compared to controls, we found modest changes in gene expression, with 42 downregulated genes and 44 upregulated genes in Mertk KO microglia (*Figure 7B, D*, *Supplementary file 5*). Despite the fact that Mertk KOs had reduced clearance of dying RGCs, there was a minimal effect on lysosomal or lipid metabolism genes when compared to GO lists (*Bult et al., 2019*). Of the 42 downregulated genes, three are involved with lysosomal processes (*Sgsh, Arsb* and *Ifi30*) and six in lipid metabolism (*Tspo, Enpp1, Ivd, Nceh1, Lbr,* and *Gpx1*). Further, when we compared our 42 downregulated genes to published datasets ATM (*Hammond et al., 2019*), PAM (*Li et al., 2019a*), DAM (*Keren-Shaul et al., 2017*), only *Nceh1* was significantly reduced (*Figure 7B, D*). Rather, *Csf1, Ccl9, Lag3,* and *Itgax* were significantly increased (*Figure 7B, D*).

Comparison of Axl KO to WT microglia revealed even fewer differentially expressed genes. Consistent with Axl being dispensable for RGC clearance, there was no change in lysosomal and lipid metabolism genes (*Bult et al., 2019*). DAM/ATM/PAM-related gene expression was largely unaltered, with the exception of *Myo1e* and *Spp1*, and other than *Spp1*, there was no change in genes associated

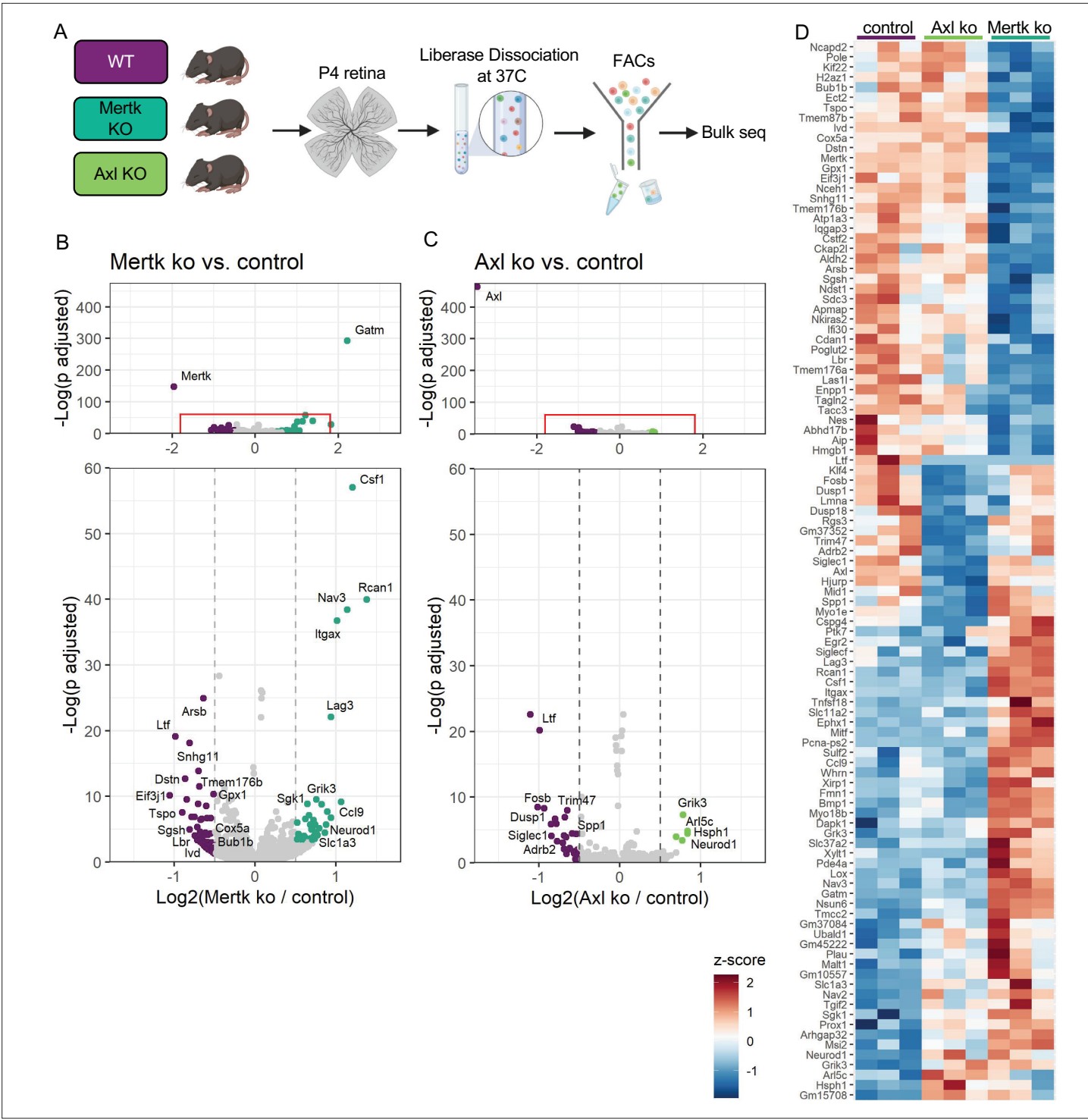

**Figure 7.** Mer and Axl are not required for expression of lysosomal, lipid metabolism, or remodeling genes. (**A**) Workflow for P4 retinal collection, dissociation, sorting, and bulk sequencing of microglia from three different groups: WT, Mertk KO, and Axl KOs (n = 3 each). (**B,C**) Volcano plot of differentially expressed genes in (**B**) Mertk KO versus WT of (**C**) Axl KO versus WT. Top, showing all genes and bottom, zoomed in to axis. Each gene is plotted according to the significance (-Log(p value)) and magnitude (Log2 (fold change)) of the difference such that those genes enriched in KO are green and those downregulated in KO are purple. Colored points indicate genes with p-value ≤ 0.05 and absolute value of Log2(fold change) > 0.5. (**D**) Heatmap of all differentially expressed genes between WT and Mertk KO and WT and Axl KO, colored by z-score of rlog values across samples.

The online version of this article includes the following figure supplement(s) for figure 7:

**Figure supplement 1.** Mertk Axl dKO microglia have a modest change in expression of lysosomal, lipid metabolism, and remodeling genes.

with survival pathways (*Figure 7C, D*, *Supplementary file 5*). The majority of differentially expressed genes were different in the two KOs (*Figure 7D*); however, sequencing of microglia from P7 Mertk/ Axl dKO retinas again revealed only a modest change in selected lysosomal, lipid metabolism (*Bult et al., 2019*), or DAM/PAM/ATM-related genes (*Figure 7—figure supplement 1A, B Supplementary file 6*). Based upon these findings, we first conclude that while phagocytosis is altered in Mertk KOs, this is not sufficient to suppress expression of most remodeling genes, suggesting that Mer-mediated signaling does not strictly drive microglia remodeling gene expression. Furthermore, while Axl facilitates microglia survival after CSF1R inhibition, it is also not required to drive microglia remodeling gene expression.

## Discussion

scRNAseq has revealed that microglia in development and disease are transcriptionally diverse across the CNS (*Masuda et al., 2020*). Due to their remarkable plasticity, a major challenge is to determine the impact of local cues on microglial state and properties. Here, using the postnatal retina as a model system, we establish neuronal cell death as a key factor driving developmental microglia diversity. We show by scRNAseq that multiple microglial states coexist within this discrete CNS region, encompassing a spectrum from homeostatic to remodeling states with high lysosomal and lipid metabolism gene expression. We establish that multiple distinct remodeling states are driven by neuronal apoptosis, and further show that several of these remodeling states survive CSF1R inhibition, while more homeostatic microglia are susceptible. CR3 and Mer are important for clearance of apoptotic retinal neurons, while Axl facilitates microglial survival following inhibition CSF1R, but neither significantly regulates expression of microglial remodeling genes. Thus, we conclude that cell death is a critical cue for driving diverse microglial properties and that multiple pathways contribute.

We identify several states of remodeling microglia characterized by reduced homeostatic gene expression and elevated lipid metabolism and lysosomal genes, with more discrete expression of various genes including chemokines/cytokines. Many of the genes in these remodeling states are shared with microglia in developing white matter tracts of the brain (CD11c/ATM/PAM) (*Wlodarczyk et al., 2017*; *Hammond et al., 2019*; *Li et al., 2019a*) and disease (DAM) (*Holtman et al., 2015*; *Keren-Shaul et al., 2017*; *Krasemann et al., 2017*). Unlike in brain, we find they represent a majority of microglia in the early postnatal retina, with most of them dependent upon neuronal apoptosis. Due to the large proportion of microglia expressing these remodeling genes, we broaden our understanding of microglia subsets present during normal development. For example, we find genes such as *Apoe* and *Ctsd* are widely expressed at varying levels, while *Fabp5* and *Spp1* are confined to more discrete subsets, suggesting that regulation of these genes and microglial states is multifaceted.

While our previous work found that microglia expressing several of these remodeling genes were dynamically regulated over development, future work analyzing the localization of these populations would further inform understanding of function and population dynamics. At present, it remains unclear whether they transition from state to state. Since we find an increase in the proportion of homeostatic microglia in Bax KO retinas, this argues that microglia in the context of neuronal death shift from a homeostatic state to more remodeling ones. We found that loss of Bax had an effect on microglial density, consistent with zebrafish studies (*Casano et al., 2016*), suggesting that neuronal apoptosis is important for regulating other microglial properties. Whether this is due to a reduction in proliferation is unknown, however we did not see a change in the proportion of cycling cells by scRNAseq.

In other contexts, efferocytosis can have a major impact on the transcriptional profile and state of the phagocyte. For example, phagocytosis promotes distinct transcriptional states in macrophages in a tissue-specific manner throughout the body (*A-Gonzalez et al., 2017*), and tissue macrophages modify their repertoire of phagocytic machinery in response to phagocytosis (*Zent and Elliott, 2017*). In the CNS, there is much to learn about how phagocytosis impacts microglial state and function (*Márquez-Ropero et al., 2020*; *VanRyzin, 2021*). One study addressing this found engulfment of newborn neurons by microglia in adult neurogenic zones stimulates a microglial secretome that subsequently regulates neurogenesis (*Diaz-Aparicio et al., 2020*). Here, in the postnatal retina, phagocytosis and digestion appear to be a major aspect of microglial heterogeneity. We postulate that some remodeling states may even represent distinct stages of the clearance and digestion process. Our analysis also identified microglial states that did not change with loss of Bax. While perhaps they

represent microglia that have not had the opportunity to interact with dying neurons, they may be involved in other important microglial functions during postnatal retinal development. For example, astrocytes are also pruned by microglia during this time period, which has been shown to be Bax-independent and also not require CR3, Mer, or CX3CR1 pathways (*Puñal et al., 2019*). Future studies could target these specific microglia subsets and try to link them to other important remodeling functions in the developing retina. Some clusters did have higher expression of ex vivo activation genes (*Marsh et al., 2020*) but the impact on developmental datasets has not been directly tested. exAM genes did not drive the formation of a single cluster and some genes were differentially expressed between samples, arguing against solely a technical effect. Further, we found exAM genes *Ccl3* and *Ccl4* were expressed in vivo, suggesting biological relevance (*Figure 1—figure supplement 5* and data not shown).

Despite the fact that postnatal RGC death has been a well-characterized model of developmental cell death of the CNS (*Péquignot et al., 2003*), the recognition pathways required for the clearance of apoptotic RGCs was previously undefined. CR3 and Mer are well-known recognition receptors in the clearance of stressed or dying neurons in various contexts (*Lemke, 2019*). Here, we show that both receptors are important for clearance of apoptotic RGCs. Previous work has suggested that complement component C1q may get deposited on exposed PtdSer (*Païdassi et al., 2008*), suggesting the pathways might work together to signal for phagocytosis, but whether complement and TAM pathways are working redundantly or cooperatively is unknown in any context. It is also unclear whether other mechanisms can compensate for the loss of either receptor. We predict this is likely, since microglia express a broad array of phagocytic receptors, and other cell types such as astrocytes can phagocytose, particularly in contexts of deficient microglia (*Puñal et al., 2019*). Furthermore, since Mer and Axl are expressed in other cell types such as astrocytes (*Fourgeaud et al., 2016*), we cannot exclude the possibility of indirect effects. While receptors Mer and Axl are both members of the TAM receptor family of tyrosine kinases (*Lemke, 2013*), it has been appreciated that Mertk is more stably expressed in microglia, while Axl is more dynamic and upregulated with inflammation or disease (*Fourgeaud et al., 2016*). Both have previously been implicated in phagocytosis and in regulating inflammatory responses (*Elliott et al., 2017*), however this is cell type-dependent (*Seitz et al., 2007*). Here, we find Mertk/Axl dKO did not have an enhanced clearance deficit for apoptotic RGCs compared to Mertk KO, although this could be due to a ceiling effect since RBPMS likely gets downregulated as apoptotic programs continue (*Elliott and Ravichandran, 2016*).

Although Mer is important for clearance, we did not see a dramatic reduction in lysosomal or lipid metabolism genes in either Mertk or Axl KO microglia. These results are consistent with disease contexts where loss of Mertk and Axl did not have a profound effect on the DAM signature in mouse model of Alzheimer's disease (*Huang et al., 2021*). This suggests that transcriptional changes in postnatal retinal microglia may depend upon other pathways, although Mer and Axl could still have essential roles in post-transcriptional regulation of microglial states. Additionally, Mertk KO microglia are likely still clearing, but at a reduced rate, since live-imaging has shown that loss of Mer results in delayed engagement with apoptotic cells but does not abolish phagocytosis completely (*Damisah et al., 2020*).

While in adulthood, retinal microglia can be ablated by blocking CSF1R signaling (*MacDonald et al., 2010*; *Anderson et al., 2019a*), we previously found that during the early postnatal period a large proportion of microglia expressing select remodeling genes are more resistant to inhibition or loss of this important survival pathway (*Anderson et al., 2019a*). One hypothesis is that specific microglial states, which are transient, have altered CSF1R dependence. Here, we identify specific remodeling states of microglia that persist following CSF1R inhibition showing reduced homeostatic gene expression. Other groups recently found small populations of CSF1R-independent or repopulating microglia in the brain, and we note similar patterns of gene expression, with reduced homeostatic genes such as *Tmem119* and *P2ry12*, and increased expression of genes such as *Lyz2* (*Zhan et al., 2020*; *Hohsfield et al., 2021*). In these microglia repopulation studies, it was proposed that more immature microglia survive CSF1R inhibition, but microglia maturation may not be a key factor in the context of development since we previously found greater dependence of retinal microglia on CSF1R at embryonic stages (*Anderson et al., 2019b*) than in postnatal retina (*Anderson et al., 2019a*). We find that most of these resistant microglia subsets are induced by neuronal apoptosis, consistent with our prior analysis showing increased microglial dependence on CSF1R in Bax KO retina (*Anderson*

*et al., 2019a*). Interestingly, in culture, peritoneal and bone marrow macrophages that phagocytose apoptotic cells also survive in the absence of serum or survival factors (*Reddy et al., 2002*). In periods of elevated cell death and possible lack of extrinsic survival factors, such as neuronally expressed CSF1R ligand, IL-34 (*Nandi et al., 2012*), phagocytes such as microglia might have evolved strategies to survive in order to maintain homeostasis or reduce inflammation (*Reddy et al., 2002*). Importantly, cholesterol, along with CSF-1/IL-34 and TGF-ß2, has been shown to be an important survival factor for microglia in culture (*Bohlen et al., 2017*). However, other possibilities exist for the overlap between clusters dependent on neuronal apoptosis (0,3,4,5) and clusters more resistant to CSF1R inhibition (3,4,5,6,10). For example, PLX-treatment itself could be inducing transcriptional changes reminiscent of remodeling states. We did see an expansion of PLX-enriched (3) microglia with the lowest levels of homeostatic gene expression and high levels of lysosomal and lipid metabolism genes. Thus, CSF1R signaling may also be important for promoting microglial homeostasis (*Hu et al., 2021*). Consistent with this, mutations in CSF1R can lead to leukoencephalopathy, a neurodegenerative disorder that is associated with loss of homeostatic microglia phenotype (*Kempthorne et al., 2020*). PLX-enriched (3) microglia may also represent cells that are undergoing other changes, even progressing toward death. It will be valuable for future studies to explore the contributions of CSF1R signaling to microglial survival and homeostasis during development.

Here, we find that Axl expression, which is regulated by neuronal apoptosis, enhances microglial survival in the absence of CSF1R signaling. Since loss of Axl did not substantially alter expression of other candidate apoptotic pathways in microglia, one possibility is that signaling directly downstream of Axl is involved in promoting survival. Axl has been shown to inhibit apoptosis in a variety of cell types (*Axelrod and Pienta, 2014*) including fibroblasts (*Goruppi et al., 1997*), oligodendrocytes in contexts of growth factor withdrawal or TNF toxicity (*Shankar et al., 2006*), and in Gonadotropin-releasing hormone neurons in the brain (*Pierce et al., 2008*). Its role in various types of cancer has been well established for promoting tumor growth and metastasis in part due to its ability to inhibit apoptosis of cancer cells (*Zhu et al., 2019*). Axl has diverse downstream signaling cascades depending on context (*Axelrod and Pienta, 2014*), but one candidate signaling cascade is the PI3K-Akt-NfkB-Bcl2 pathway. Activation of PI3K and Akt has shown to be important for survival of cultured macrophages (*Reddy et al., 2002*), oligodendrocytes (*Shankar et al., 2003*; *Shankar et al., 2006*), neurons (*Pierce et al., 2008*) as well as cancer cells (*Axelrod and Pienta, 2014*). Another candidate pathway for regulating microglial survival is Spp1/osteopontin since we found it was significantly downregulated in Axl KO microglia. Spp1/osteopontin has been shown to promote survival of other cell types including peripheral immune cells (*Denhardt et al., 2001*) and cancer cells (*Saleh et al., 2016*). Furthermore, Spp1/osteopontin can influence aspects of microglial properties including enhancing survival under conditions of stress in culture (*Yu et al., 2017*).

Altogether, we show neuronal cell death is a key factor driving multiple states of microglia in developing retina, with distinct transcriptional profiles and altered dependence from CSF1R signaling for survival. Apoptotic cell recognition by microglia is thus a critical developmental event driving diverse effects on gene expression, clearance, and survival, with distinct pathways involved in mediating each of these responses.

# Materials and methods

## Key resources table

| Reagent type (species) or resource | Designation | Source or reference | Identifiers | Additional information |
|---|---|---|---|---|
| Strain, strain background (*Mus musculus*, M/F) | B6.129P2 (Cg)-*Cx3cr1*tm1Litt/J | Jackson Laboratories *Jung et al., 2000* | 005582 | A kind gift from Dr. Richard Lang with permission from Dr. Steffen Jung |
| Strain, strain background (*Mus musculus*, M/F) | B6.129 × 1-*Bax*tm1Sjk/J | Jackson Laboratories *Knudson et al., 1995* | 002994 | |
| Strain, strain background (*Mus musculus*, M/F) | B6.129S4-*Itgam*tm1Myd/J | Jackson Laboratories *Coxon et al., 1996* | 003991 | |

*Continued on next page*

*Continued*

| Reagent type (species) or resource | Designation | Source or reference | Identifiers | Additional information |
|---|---|---|---|---|
| Strain, strain background (*Mus musculus*, M/F) | B6.129-*Mertk*$^{tm1Grl}$/J | **Lu et al., 1999** | | A kind gift from Dr. Greg Lemke |
| Strain, strain background (*Mus musculus*, M/F) | B6.129-*Axl*$^{tm1Grl}$/J | **Lu et al., 1999** | | A kind gift from Dr. Greg Lemke |
| Antibody | (Goat polyclonal) anti-GFP | Abcam | Cat# ab5450 RRID: AB_304897 | IF (1:2000) |
| Antibody | (Rabbit monoclonal) anti-C1q | Abcam | Cat# ab182451 RRID: AB_2732849 | IF (1:1500) |
| Antibody | (Rabbit monoclonal) anti-active caspase-3 | BD Biosciences | Cat# 559,565 RRID: AB_397274 | IF (1:500) |
| Antibody | (Guinea pig polyclonal) anti-RBPMS | Millipore Sigma | Cat# ABN1376 RRID: AB_2687403 | IF (1:1000) |
| Antibody | 488 (Donkey polyclonal) anti-goat | Invitrogen | Cat# A11055 RRID: AB_2534102 | IF (1:400) |
| Antibody | 555 (Donkey polyclonal) anti-rabbit | Thermo Fisher Scientific | Cat# A31572 RRID: AB_162543 | IF (1:400) |
| Antibody | 647 (Donkey polyclonal) anti-guinea pig | Jackson ImmunoResearch | Cat# 706-605-148 RRID: AB_2340476 | IF (1:400) |
| Antibody | BV421 (Rat monoclonal) anti-CD45 | BD Bioscience | Cat# 563,890 RRID: AB_2651151 | FACS (1:200) |
| Antibody | 488 (Rat monoclonal) anti-CD11b | BD Bioscience | Cat# 557,672 RRID: AB_396784 | FACS (1:200) |
| Antibody | PE (Rat monoclonal) anti-CCR2 | R&D Systems | Cat# FAB5538P RRID: AB_10718414 | FACS (1:200) |
| Antibody | APC (Rat monoclonal) anti-Ly6C | BD Bioscience | Cat# 560,595 RRID: AB_1727554 | FACS (1:200) |
| Recombinant DNA reagent | Cx3cr1 ISH probe | Molecular Instruments, **Choi et al., 2018** | | |
| Recombinant DNA reagent | Ccl3 ISH probe | Molecular Instruments, **Choi et al., 2018** | | |
| Peptide, recombinant protein | FITC IB4-lectin | Sigma-Aldrich | Cat# L9381 | IF (1:400) |
| Commercial assay or kit | In situ hybridization chain reaction v3.0 (HCR) | Molecular Instruments (Los Angeles, CA) **Choi et al., 2018** | | |
| Commercial assay or kit | RNeasy Plus Micro Kit | Qiagen | Cat# 74,034 | |
| Commercial assay or kit | NEBNext rRNA Depletion Kit (human/mouse/rat) | New England BioLabs | Cat# E6310L | |
| Commercial assay or kit | NEBNext Ultra II RNA Library Prep Kit for Illumina | New England BioLabs | Cat# E7770L | |
| Commercial assay or kit | Agilent D1000 ScreenTape assay | Agilent | Cat# 5067–5582 and 5067–5583 | |
| Commercial assay or kit | Kapa Biosystems Kapa Library Quantification Kit | Roche | Cat# KK4824 | |
| Commercial assay or kit | NovaSeq XP kit v1.5 | Illumina | Cat# 20043131 | |
| Commercial assay or kit | NovaSeq 6,000 S4 reagent kit v1.5 | Illumina | Cat# 20028312 | |
| Chemical compound, drug | Pexidartinib (PLX3397) | AdooQ BioScience | Cat# A15520 | |
| Chemical compound, drug | corn oil | Sigma-Aldrich | Cat# C8267 | |
| Chemical compound, drug | DMSO | Fisher Scientific | Cat# BP231 | |
| Chemical compound, drug | Liberase TM | Sigma-Aldrich | Cat# 5401119001 | |
| Chemical compound, drug | Red Blood Cell Lysis Buffer | eBioscience | Cat# 00-4333-57 | |

*Continued on next page*

*Continued*

| Reagent type (species) or resource | Designation | Source or reference | Identifiers | Additional information |
|---|---|---|---|---|
| Chemical compound, drug | Mouse Fc Block | BD Biosciences | Cat# 553,142 | |
| Chemical compound, drug | DNase I | Sigma-Aldrich | Cat# D4513 | |
| Chemical compound, drug | Fluoroshield mounting medium with DAPI | Millipore Sigma | Cat# F6057 | |
| Software, algorithm | Biorender | Biorender, Toronto, ON | | |
| Software, algorithm | Nikon Elements | Nikon, Melville, NY | | |
| Software, algorithm | Prism (v9.0) | GraphPad, La Jolla, CA | | |
| Software, algorithm | FlowJo software | Flowjo, LLC, Ashland, Oregon | | |
| Commercial assay or kit | Chromium Single Cell 3' GEM, Library & Gel Bead Kit v3 | 10 X Genomics | PN-1000075 | |
| Software, algorithm | cellranger (v3.1.0) | 10 X Genomics | | |
| Software, algorithm | Seurat (v3.1.5) | *Stuart et al., 2019* | | |
| Software, algorithm | scSplit (v1.0.0) | *Xu et al., 2019* | | |
| Software, algorithm | samtools view (v1.8) | *Danecek et al., 2021* | | |
| Software, algorithm | freebayes (v1.3.1) | *Garrison and Marth, 2012* | | |
| Software, algorithm | Vcffilter (v1.0.1) | *Garrison et al., 2021* | | |
| Software, algorithm | Bcftools merge (v1.9) | *Danecek et al., 2021* | | |
| Software, algorithm | BBmap (v38.34) | Bushnell B. http://sourceforge.net/projects/bbmap | | |
| Software, algorithm | cutadapt (v1.16) | *Martin, 2011* | | |
| Software, algorithm | STAR (v2.7.9a) | *Dobin et al., 2013* | | |
| Software, algorithm | featureCounts (v1.6.3) | *Liao et al., 2014* | | |
| Software, algorithm | DESeq2 (v1.32.0) | *Love et al., 2014* | | |

## Experimental model and subject details

### Animal husbandry and procedures

All animals were treated within the guidelines of the University of Utah Institutional Animal Care and Use Committee (IACUC) and all experiments were IACUC approved. Mice were housed in an AAALAC accredited animal facility with 12 hr light/12 hr dark cycles and ad libitum access to food and water. Both sexes were used for all experiments. Information on the ages of mice used for each experiment can be found in the figures/text. Pexidartinib (PLX3397) was dissolved in corn oil and 10% DMSO and administered to postnatal pups by daily intraperitoneal injection P3-P5 or P4-P6 at 0.25 mg/g body weight. Mice were euthanized by isoflurane asphyxiation followed by decapitation.

### Mouse strains

The B6.129P2 (Cg)-*Cx3cr1*$^{tm1Litt}$/J mice were a gift from Richard Lang with permission from Dr. Steffen Jung (*Jung et al., 2000*). B6.129 × 1-*Bax*$^{tm1Sjk}$/J mice (JAX 002994) (*Knudson et al., 1995*) and B6.129S4-*Itgam*$^{tm1Myd}$/J (JAX 003991) (*Coxon et al., 1996*) were purchased from the Jackson Laboratory and both were crossed with the B6.129P2 (Cg)-*Cx3cr1*$^{tm1Litt}$/J strain. The B6.129-*Mertk*$^{tm1Grl}$/J and B6.129-*Axl*$^{tm1Grl}$/J strains (*Lu et al., 1999*) were a kind gift from Dr. Greg Lemke and double knockouts were generated in house. Analyses were performed prior to rod photoreceptor degeneration in Mertk KO (*Duncan et al., 2003*).

### Tissue processing

Following euthanasia, retinas were dissected in ice-cold 0.1 M PBS. For retinal whole mounts for immunostaining, eyes were removed from the head and retinas were carefully dissected from the rest of the eye (cornea, lens, RPE, hyaloid vasculature, vitreous, ciliary body) in ice-cold PBS. Whole neural

retinas were washed in PBS for 10–20 min and then fixed in 4% PFA for 30–45 min at room temperature. For RNAase-free dissections for qHCR or FACs, retinas were carefully dissected in RNase-free conditions using ice-cold, sterile RNase-free PBS, removing all non-neural eye tissue (ciliary body, pigmented epithelium, vitreous).

## Immunohistochemistry

As previously done (*Anderson et al., 2019a*), whole retinas were fixed in 4% PFA for 30–40 min and then washed in ice-cold PBS three times for 5–10 min each. Retinas were incubated for 1 hr at room temperature in blocking buffer (0.2% triton-X, 10% BSA, 10% normal donkey serum in 0.01 M PBS), and subsequently incubated in primary antibody for 2 days at 4 °C in (0.2% triton-X, 5% BSA in 0.01 M PBS). They were then washed three times with PBS and incubated in secondary antibodies (5% BSA in PBS) for 2 hr at room temperature, washed, and mounted with Fluoroshield mounting medium with DAPI. Antibody information in Key Resources Table.

## In situ hybridization chain reaction (HCR)

As previously done (*Anderson et al., 2019a*), wholemount retinas were fixed overnight in 4% PFA in 4 °C. Retinas were washed and dehydrated in methanol/PBS at 25%, 50%, two times 100% for 15 min each, stored in 100% methanol overnight at 4 and 20°C long term. In situ hybridization was performed as published using v3.0 reagents from Molecular Instruments (https://www.molecularinstruments.com) (*Choi et al., 2018*). Briefly, samples were rehydrated using (75% Methanol/25% PBST, 50% Methanol/50% PBST, 25%, Methanol/75% PBST, two times 100% PBST), treated with Proteinase K, and post fixed 20 min at room temperature in 4% PFA, and washed three times with PBST. Pre-hybridization was performed in 30% probe hybridization buffer for 30 min at 37 °C, and retinas were placed in hybridization buffer at 37 °C overnight. Retinas were washed, placed in amplification buffer for 30 min at room temperature. Separately, hairpins used for amplification were denatured at 95 °C for 90 s and cooled to room temperature for 30 min. Retinas were placed in amplification buffer with hairpins at room temperature in the dark overnight. Retinas were washed, DAPI stained, and mounted on slides. Probes recognizing all known transcript variants for each of Cx3cr1 and Ccl3 were generated by Molecular Instruments. Using the B6.129P2 (Cg)-*Cx3cr1*[tm1Litt]/J (*Jung et al., 2000*), the fluorescent signal represents detection of Cx3cr1 mRNA by HCR and GFP fluorescence which persists through the procedure.

## Confocal microscopy

Confocal images were acquired on an inverted Nikon A1R Confocal Microscope. Images were acquired at 20 X objective with a 3 X digital zoom. Multi-points were stitched with a 10% overlap. Images of retinal whole mounts were 144 multi-point images (on average) to obtain the entire dorsal retina. Stacks through the Z plane were at 0.8 µm steps of about ~13 µm thickness to capture just the nerve fiber layer (NFL) and ganglion cell layer (GCL) at 0.2 µm pixel resolution. Whole mount retina images represent max projections of inner retina (NFL/GCL) from the central retina (optic nerve) to the periphery (edge of retina). In cases of microglial quantification, whole mount imaging and analysis spanned ~25 µm thickness from NFL to inner plexiform layer (IPL). Image acquisition settings were consistent across ages and genotypes.

## Dissociation, fluorescence-activated cell sorting (FACS) and flow cytometry

Except for SC-sequencing, we pooled two retinas from an individual animal for each sample for flow cytometry and FACs. Freshly dissected pure retinas were dissociated in PBS, 50 mM HEPES, 0.05 mg/ml DNase I, and 0.025 mg/ml Liberase for 35 min with intermediate trituration at 37 °C. Cells were passed through a 70 µm nylon cell strainer, washed with ice-cold staining buffer (1 X PBS, 2% BSA, 0.1% sodium azide, and 0.05% EDTA), and red blood cells were lysed. Cell counts were determined using a cell counter (Invitrogen Countess) and Fc block was added at 2 µL per $10^6$ cells. Antibodies were applied for 30 min on ice. Antibody information in Key Resources Table. Cells were washed, pelleted, and resuspended in 500 µL staining buffer. FACS was performed using a BD FACS Aria cell sorter at the University of Utah Flow Cytometry Core. Forward and side scatter were used to eliminate debris, and both the width and area of the forward and side scatter was used to discriminate singlets.

For flow analysis, roughly 1 million singlet events (300,000 in rare cases) were recorded for flow analysis using FlowJo software (Flowjo, LLC, Ashland, Oregon).

## Single-cell RNA sequencing

To ensure we captured enough cells for analysis (*Liddelow et al., 2020*), we pooled 13 animals/26 retinas for each Bax WT (Sample 1) and littermate KO (Sample 2) from six litters. 12 animals/24 retinas for PLX3397 CX3CR1-GFP/ + and 11 animals/22 retinas were pooled for Vehicle CX3CR1-GFP/ + controls each from two litters. Altogether, sequencing data represent 49 animals divided into two experiments: Bax/WT and PLX/Veh. The dissociation was performed as normal, except pooling six retinas per tube during dissociation. CD45$^+$ CD11b$^+$/GFP$^+$ CCR2$^-$ cells were sorted for Bax samples and CD45$^+$GFP$^+$ Ly6C$^-$ cells were sorted for PLX and Vehicle samples. Cells were sorted using a 5 laser BD FACSAria with a 70 µm nozzle at 55 psi by the University of Utah Flow Core into a cold, empty tube before library generation. 15,000 cells for each Bax sample and 7000–10,000 cells for PLX and Vehicle were loaded onto 10 X chip. Single-cell libraries were generated with 10 X Genomics Single Cell 3′ Gene Expression Library Prep v3 reagents at the University of Utah High Throughput Genomics core. The libraries were sequenced on a NovaSeq 6000 to generate at least 200 million paired end reads of 150 bp per sample.

## Analysis of single-cell RNA-seq data

Aligned reads to the mm10 reference from 10 X genomics (version 3.0.0 from Ensembl 93) and generated feature-barcode matrices using cellranger count v3.1.0 with expected-cells set to 5000. Filtered feature matrices from cellranger were further filtered with Seurat (v3.1.5) (*Stuart et al., 2019*). High-quality cells were selected by <10% mitochondrial gene content, and high number of features (genes). Feature cut-off was specific to each sample: > 1500 for Sample 1, > 1000 for Sample 2, > 1800 for vehicle, and >2500 for PLX-treated. Likely doublets, with high transcript counts, were also eliminated: < 27,000 counts for Sample 1, < 25,000 for Sample 2, < 47,000 for vehicle, and <125,000 for PLX treated. In silico genotyping, as described below, was used to assign cells to the correct Bax groups. Filtered cells from all samples were then combined, without batch correction, using Seurat's merge function and processed with the SCTransform pipeline (*Hafemeister and Satija, 2019*). Mitochondrial percentage was regressed out of the model and Bax and presumed Bax-linked genes (Ftl1 and Ftl1-ps1) were excluded from the variable features list used for dimensionality reduction. Ftl1 and Ftl1-ps1 are dramatically regulated by Bax genotype. The Ftl1 gene locus is adjacent to the Bax gene locus and we expect that the Bax mutation impacts Ftl1 and Ftl1-ps1 regulation. We have no reason to believe that this has functional consequences, so we chose to mitigate this source of variation.

From here, we subsetted the data to include only microglial cells. We identified non-microglia populations by their expression of Ptprc, Plac8, Clec12a, Ms4a7, Mrc1, and Rorb, and reran the SCTransform pipeline as before with the exception that FindClusters was run with resolution set to 0.4. Comparison between treatments, clusters, or type (homeostatic vs remodeling) were made with differentially expressed genes, identified with the FindAllMarkers function which, by default, uses natural-log normalized counts (log1p(counts)) for this analysis.

## In silico genotyping

During single-cell library preparation, cells from one Bax WT and one Bax KO animal were sorted into the wrong sample so we chose to call these mixed samples Sample1 and Sample2. We used the scSplit pipeline (*Xu et al., 2019*) to perform in silico genotyping and reassign these cells to the correct condition. Our rationale was that the only thing that would be different between Bax KO and Bax WT littermates would be the Bax gene. As Bax expression is sparse in our dataset, we examined Chromosome 7 (where the Bax locus resides) to identify single nucleotide polymorphisms (SNPs) that would segregate with one of the two Bax alleles. These SNPs were identified within the Sample 1 and Sample 2 genome-aligned reads, without regard to the individual cells, then scSplit created a matrix, comparing SNPs to cell barcodes, to define two groups of cells with distinct SNP profiles. From there we defined which of the two groups was Bax KO by lower average expression of Bax. We acknowledge that the correct genotype cannot be determined with perfect accuracy by this method, but we find that the minority of cells are re-assigned, as we would expect, and that cells cluster more tightly by genotype than by sample.

More specifically, beginning with the bam genome alignment file for each sample generated by cellranger, we used samtools view v1.8 (*Danecek et al., 2021*) to exclude reads with mapping quality <10, or flagged as unmapped, not primary alignment, fails quality checks, PCR or optical duplicate, or supplementary alignment. These filtered bam files were used to identify SNPs with freebayes v1.3.1 (*Garrison and Marth, 2012*). Indels, MNPs, and complex alleles were filtered from the input to the algorithm. Region was set to chromosome 7, minimum allele count was set to 2, use-best-n-alleles was set to 2, and minimum base quality was set to 1. Vcffilter v1.0.1 (*Garrison et al., 2021*) was used to exclude SNPs with quality score $\leq$ 30. We found that some SNPs were identified in one sample and not the other, so Bcftools merge v1.9 (*Danecek et al., 2021*) was used to combine all SNPs into a single file. ScSplit count v1.0.0 (*Xu et al., 2019*) was used to the combined SNPs to generate count matrices for the filtered cell barcodes identified by cellranger, and scSplit run, with expected number of mixed samples set to 1, genotyped the individual cells based on these matrices. The improvement in sample separation was assessed with UMAP reduction through the SCTransform pipeline using default settings and dimensions 1:30.

## Bulk RNA sequencing

For P4 WT, Mertk KO, and Axl KO samples, CD11b$^+$ CD45$^+$ Ly6c$^-$ cells from one animal (two retinas) were sorted directly into RLT buffer (from Qiagen RNeasy MicroRNA kit 74034) using a 4 laser BD FACSAria at the University of Utah Flow Core and stored at –80 °C. 2 samples were pooled prior to RNA isolation so that each replicate was two animals/four retinas. WT samples had 5470, 6134, and 7029 cells, Mertk KO samples had 4245, 4038, and 8591 cells, and AXL KO samples had 5790, 5282, and 5071 cells. For P7 WT and Mertk/Axl dKO, CD11b$^+$ CD45$^+$ cells from one animal (two retinas) were sorted directly into RLT buffer and stored at –80 °C. P7 samples were not pooled and thus represent one animal/two retinas. For WT samples, 3783, 4083, and 4982 cells were collected. For Mertk/Axl dKO samples, 2408, 2633, and 2388 cells were collected. We sequenced RNA from three samples for each genotype in a single experiment.

RNA from all samples was purified using the RNeasy Plus Micro kit. The University of Utah High-throughput Genomics core then hybridized total RNA with the NEBNext rRNA Depletion Solution human/mouse/rat to substantially diminish cytoplasmic and mitochondrial rRNA from the samples. Stranded RNA sequencing libraries were prepared as described using the NEBNext Ultra II RNA Library Prep Kit for Illumina. Purified libraries were qualified on an Agilent Technologies 2200 TapeStation using a D1000 ScreenTape assay. The molarity of adapter-modified molecules was defined by quantitative PCR using the Kapa Biosystems Kapa Library Quantification Kit. Individual libraries were normalized to 10 nM, and equal volumes were pooled in preparation for Illumina sequence analysis. Sequencing libraries were chemically denatured and applied to an Illumina NovaSeq flow cell using the NovaSeq XP workflow. Following transfer of the flowcell to an Illumina NovaSeq 6000 instrument, a 150 × 150 cycle paired end sequence run was performed using a NovaSeq 6000 S4 reagent Kit v1.5. Samples were sequenced to a depth of 39–76 million reads.

## Analysis of bulk RNA-seq data

Optical duplicates were removed with clumpify using BBmap v38.34 (Bushnell B. http://sourceforge.net/projects/bbmap) and default settings, then Illumina adapters were trimmed with cutadapt v1.16 (*Martin, 2011*) using a minimum overlap of 6 and minimum length of 20. Alignment to mm10/Ensembl release 102 (P4 samples) or mm39/Ensembl release 104 (P7 samples) was accomplished with STAR v2.7.9a (*Dobin et al., 2013*) using mouse Ensembl release 104 with overhang set to 124. Trimmed reads were aligned in two pass mode to generate a BAM file, sorted by coordinates. Reads were assigned to the target with the largest overlap, and uniquely aligned, reversely stranded reads were counted with featureCounts v1.6.3 (*Liao et al., 2014*). Differentially expressed genes were identified from counts using DESeq2 v1.32.0 (*Love et al., 2014*). Features with fewer than five reads in every sample were eliminated before DESeq was run. rlog-transformed values were used for sample visualizations.

## Comparisons to other datasets

We used the following gene lists for comparison to our single-cell and bulk sequencing datasets. DAM genes were defined as the top 150 differentially expressed genes when comparing DAM to

homeostatic microglia from Table S3 of *Keren-Shaul et al., 2017*. For developmental gene lists, we used all cluster one PAM markers from Table S1 of *Li et al., 2019a* and cluster four ATM markers from Table S1 of *Hammond et al., 2019*. For lysosomal and lipid metabolism genes we used GO lists of 552 'Lysosomal' genes (GO:0005764) and 1428 'Lipid Metabolic Process' genes (GO:0006629) from the MGI mouse genome database (MGD) (*Bult et al., 2019*). For comparison to interferon-responsive microglia, we used cluster eight markers from Table 5 of *Dorman et al., 2022*. For exAM activation genes, we used 27 genes from Table 4 of *Marsh et al., 2020*.

## Quantification and statistical analysis

### Image analysis

All counts were performed blinded and manual, using Nikon Elements software (Melville, NY).

For double-positive CC3$^+$RBPMS$^+$ or single RBPMS$^+$ counts of retinal whole mounts, two ROIs of roughly 0.4 mm$^2$ of central and periphery of dorsal retina were analyzed and then averaged. Images were max projected and roughly 13 µm thick, spanning the NFL to the GCL. For CC3$^+$ density, the entire dorsal leaf was analyzed, roughly 2–3 mm$^2$ again spanning the NFL to GCL only. See also *Figure 5—figure supplement 2*. For microglial quantification, whole mount imaging and analysis spanned ~25 µm thickness from NFL to IPL and 0.5625 mm$^2$ of the central to mid-peripheral, vascularized retina of the dorsal leaf.

### Statistical methods

Detailed statistical information can be found in the graphs, figure legends and *Supplementary file 7* including tests used, sample size, and precision measures. For image and flow analysis, a minimum of 4 samples (biological replicates), collected from ≥2 litters (technical replicates), were obtained for each genotype. All image and flow data were analyzed using Prism 9 software (GraphPad, La Jolla, CA). All data were first tested for normality using four different tests: Anderson-Darling, D'Agostino & Pearson, Shapiro-Wilk, and Kolmogorov-Smirnov test. If any one test failed, non-parametric tests were used. We tested for heteroscedasticity in groups of three or more by a Brown-Forsythe test. If not significantly different, we ran an Ordinary one-way ANOVA with post-hoc Tukey's multiple comparison test. If the standard deviations were significantly different between groups, we ran a Welch's ANOVA with post-hoc Dunnett's T3 multiple comparison's test. Outliers were not excluded. For all data that is presented as the mean, error bars indicate the standard error of the mean, SEM. We used a 95% confidence interval and a p-value of <0.05 for rejecting the null hypothesis. For image and flow data, exact p-values are reported in figure legends and *Supplementary file 7*. For sequencing data, exact p-values are reported in corresponding tables. Levels of significance were represented as \*p < 0.05, \*\*p < 0.01, \*\*\*p < 0.001, \*\*\*\*p < 0.0001 unless otherwise specified and for image and flow data, represent multiple comparison test results.

## Acknowledgements

This research was supported by the National Eye Institute and the National Institute of Neurological Disorders and Stroke of the National Institutes of Health under Awards R01EY030307 (MLV), and T32EY024234, T32NS115664 (NG). We thank the University of Utah BPRB Animal Facility. Cartoons created with BioRender. Research reported in this publication utilized the University of Utah Flow Cytometry Core, High-Throughput Genomics and Bioinformatics Analysis Shared Resource at Huntsman Cancer Institute at the University of Utah and was supported by the National Cancer Institute of the National Institutes of Health under Award Number P30CA042014. The content is solely the responsibility of the authors and does not necessarily represent the official views of the NIH. The computational resources used were partially funded by the NIH Shared Instrumentation Grant 1S10OD021644-01A1.

# Additional information

## Funding

| Funder | Grant reference number | Author |
|---|---|---|
| National Eye Institute | R01EY030307 | Monica L Vetter |
| National Eye Institute | T32EY024234 | Nathaniel Ghena |
| National Institute of Neurological Disorders and Stroke | T32NS115664 | Nathaniel Ghena |

The funders had no role in study design, data collection and interpretation, or the decision to submit the work for publication.

## Author contributions

Sarah Rose Anderson, Conceptualization, Formal analysis, Investigation, Methodology, Supervision, Validation, Visualization, Writing - original draft; Jacqueline M Roberts, Data curation, Formal analysis, Investigation, Methodology, Software, Visualization, Writing – review and editing; Nathaniel Ghena, Formal analysis, Investigation, Visualization, Writing – review and editing; Emmalyn A Irvin, Joon Schwakopf, Isabelle B Cooperstein, Formal analysis, Investigation; Alejandra Bosco, Formal analysis, Investigation, Writing – review and editing; Monica L Vetter, Conceptualization, Funding acquisition, Project administration, Supervision, Writing – review and editing

## Author ORCIDs

Sarah Rose Anderson http://orcid.org/0000-0002-7639-4064
Jacqueline M Roberts http://orcid.org/0000-0002-1024-3976
Nathaniel Ghena http://orcid.org/0000-0003-1890-8036
Monica L Vetter http://orcid.org/0000-0001-6262-8504

## Ethics

This study was performed in strict accordance with the recommendations in the Guide for the Care and Use of Laboratory Animals of the National Institutes of Health. All of the animals were handled according to approved institutional animal care and use committee (IACUC) protocols (#18-08013 and #21-08001) of the University of Utah.

## Decision letter and Author response

Decision letter https://doi.org/10.7554/eLife.76564.sa1
Author response https://doi.org/10.7554/eLife.76564.sa2

---

# Additional files

## Supplementary files

• Supplementary file 1. scRNAseq cluster markers. Upregulated and downregulated genes for each of the 11 scRNAseq clusters.

• Supplementary file 2. Homeostatic versus remodeling cluster comparison. Differentially expressed genes in homeostatic clusters (1,2) compared to remodeling clusters (0,3,4,5,6,10).

• Supplementary file 3. Bax KO vs WT cell comparison. Differentially expressed genes in Bax KO cells compared to Bax WT cells.

• Supplementary file 4. PLX vs Vehicle cell comparison. Differentially expressed genes in PLX cells compared to Vehicle cells.

• Supplementary file 5. Axl and Mertk KO microglia DESeq. Differentially expressed genes from bulk RNA-seq of P4 Mertk KO and Axl KO microglia compared to age-matched wildtype controls.

• Supplementary file 6. Axl Mertk dKO microglia DESeq. Differentially expressed genes from bulk RNA-seq of P7 Mertk Axl dKO microglia compared to age-matched wildtype controls.

• Supplementary file 7. Statistics. Statistics for *Figure 5*, *Figure 5—figure supplement 1*, and *Figure 6*.

• Transparent reporting form

## Data availability

Sequencing data have been deposited in GEO under the reference series GSE192602. https://www.ncbi.nlm.nih.gov/geo/query/acc.cgi?acc=GSE192602 Data generated or analyzed during this study are included in the manuscript.

The following dataset was generated:

| Author(s) | Year | Dataset title | Dataset URL | Database and Identifier |
|---|---|---|---|---|
| Vetter ML | 2022 | Neuronal apoptosis drives remodeling states of microglia and shifts in survival pathway dependence | https://www.ncbi.nlm.nih.gov/geo/query/acc.cgi?acc=GSE192602 | NCBI Gene Expression Omnibus, GSE192602 |

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
