## [Editor Report]

Your study highlights important and novel functions of neuronal apoptosis in shaping microglial states in the retina, as well as in the characterization of the underlying pathways. This work not only expands our understanding of core features of neural development but more broadly our comprehension of microglial biology across physiological and pathophysiological conditions. It will be of great interest to the field and the broad readership of *eLife*.

---

## [Decision Letter]

**Decision letter after peer review:**

Thank you for submitting your article "Neuronal apoptosis drives remodeling states of microglia and shifts in survival pathway dependence" for consideration by *eLife*. Your article has been reviewed by 3 peer reviewers, and the evaluation has been overseen by a Reviewing Editor and Marianne Bronner as the Senior Editor. The reviewers have opted to remain anonymous.

Your study highlights important and novel functions of cell death in shaping microglial states in the retina, as well as the pathways involved. However, the three reviewers have raised some concerns that you will find detailed below. In particular, they raised some points that should addressed in your revised manuscript:

1) The single cell transcriptomic results appear to have been performed on a single biological replicate, which makes it difficult to convincingly draw strong conclusions especially on the relative shifts in population sizes. To address these concerns, it would be important to either add a biological replicate of the transcriptomic study or to confirm by distinct techniques (such as in vivo staining) that the different microglial states are reduced or increased in different experimental conditions examined by the authors.

2) Several points were raised regarding a need for clarification of the methodological approaches used in the transcriptomic analyses that should be addressed to strengthen the conclusions put forward as well as to provide some of the lists of genes discussed by the reviewers. It would be helpful for the reader to follow the flow of the analyses and manuscript.

3) While the link between CSF1R resistance and the induction of a "remodelling state" could be supported by additional experiments, we believe that this lies outside of the scope of this already important study. Nonetheless, it would be important for the authors to down-tone this specific conclusion and discuss additionnal possibilities.

*Reviewer #1 (Recommendations for the authors):*

Reviewer concerns and suggestions for improvement:

1. Additional methodological details about scRNAseq processing are needed.

First and most importantly: In line 566 (Methods) the authors state that filtered cells from all 4 datasets were merged and then processed with the [Seurat] SCTransform pipeline. Were the 4 datasets combined using Seurat's integration/batch correction functionality, or were they simply merged? If the former, please provide in the Methods the details of the integration workflow. If the latter, the authors should explain the steps that were taken, if any, to mitigate batch effects. In the absence of such steps, there is a concern that batch effects, rather than true biological differences, could be the major driver of the differences in cluster composition between the WT/KO/vehicle/PLX samples.

2. Second, the authors do not state which count data from the Seurat data object were used to make figures and perform differential expression analysis. The standard in the field is to use log-normalized read counts. But since the SCTransform pipeline was used, the object will also contain "corrected" counts obtained via modeling. The methods should state which were used. If the latter were used the authors should justify this. It is particularly important to consider the appropriate methods for identifying differentially expressed genes (DEGs) because cluster 0, the major remodeling cluster, only had 4 upregulated DEGs that distinguished it from other clusters (Table S1). Two of these, Apoe and Ctsd, are important to the authors argument that cluster 0 cells have a remodeling phenotype. Further, some studies use lack of upregulated DEGs as an argument for "overclustering" i.e. that the clustering algorithm has split the data in a non-meaningful way. So a finding of zero DEGs would be significant for interpretation. For these two reasons it is critical to establish that the cluster 0 markers were identified in a sound manner.

3. Because Cluster 0 lacks definitive markers (Table S1), the reader is left wondering why exactly the clustering algorithm defined it as distinct from other clusters. The clear implication of the text is that cluster 0 is mainly defined by downregulation of homeostatic (cluster 1) markers. Four qualitative examples of this are given in Figure 2b. This point should be supported by a more systematic analysis. There are a number of ways this could be done, e.g. by showing DEGs downregulated in cluster 0 (compared to cluster 1 or to the entire dataset), or by plotting module scores for a larger set of homeostatic genes. Such an analysis is important to establish what genetic differences distinguish the cluster 0 cells, and whether they are biologically plausible or instead might indicate an overclustering artifact.

4. In the section on bulk RNAseq of Mer and Axl mutants, it is stated that there was no effect on lysosomal or lipid metabolism genes or DAM/ATM/PAM genes (lines 310-12 and 318-19). This is not clearly demonstrated by the data in Figure 7. It is great to see the heatmap listing all differentially expressed genes, but the authors should clarify that this list is consistent with the statements in the main text. (The reader cannot be expected to know what all of these genes do!) This could be done either by a gene ontology analysis or by explicitly providing a list of lysosomal and lipid metabolism genes that was used to compare with the differentially expressed gene list from Figure 7D. As for the DAM/ATM/PAM genes -- some are highlighted in the figures and text so this category is better supported. But I still think it would improve clarity and methodological transparency if the authors would provide a list of the genes they defined as DAM/ATM/PAM for this and other analyses used throughout the study.

5. Other scRNAseq studies using enzymatic dissociation have found clusters of ex-vivo activated microglia. See Marsh et al. 2020 biorxiv (cited by the authors). The fact that cells expressing this distinctive profile are missing is noteworthy and merits explanation. Were these cells detected and filtered out? Were transcriptional inhibitors used?

6. The conceptual explanation for the Bax in silico genotyping strategy needs to be expanded, either in the supplementary figure legend or Methods. I appreciated the supplementary figure, but I still don't really understand what the overall strategy was or what the plots in Figure S3C-F are showing. Why was it expected that BaxWT and KO would have distinct SNPs? How were the SNPs identified (at a conceptual level; the explanation of software tools is reasonably clear)? Why are Ftl1 and Ftl-ps1 part of this analysis?

7. The plots in Figure 2B-E (UMAP Feature Plots) give a qualitative sense of how these genes change across clusters. The authors' argument would be strengthened by pairing this with a more quantitative view of how gene expression differs between homeostatic and remodeling groups. E.g. by using dot plots, violin plots, or explicit tests for up/downregulated genes comparing of homeostatic vs. remodeling groups.

8. The analysis of apoptotic clearance in Mer and Axl mutants is compelling. However only one time point, near the end of the death period, was checked (P5). Do apoptotic bodies persist in these mutants or can other mechanisms compensate for the loss of Mer/Axl?

*Reviewer #2 (Recommendations for the authors):*

In the article titled "Neuronal apoptosis drives remodeling states of microglia and shifts in survival pathway dependence" the authors describe the heterogeneity of the microglial population in the early postnatal retina. In WT mice, the authors find both "homeostatic" microglia (expressing Tmem119 and P2RY12) and several clusters of "remodeling" microglia (expressing lysosomal and phagocytic components). In a Bax KO mouse lacking the significant apoptosis that usually occurs in the early postnatal retina, they find a decrease in the remodeling clusters of microglia, concluding that the lack of apoptosis in this mouse prevents the shift in microglial phenotype from homeostatic to remodeling. Several of these "remodeling" clusters in WT mice are relatively enriched after short term PLX3397 treatment, a CSF1R inhibitor commonly used to deplete microglia in mouse.

This paper provides interesting new phenotyping of retinal microglia and how they change with Bax KO and PLX treatment, but one central conclusion is currently not fully supported by the data provided. The central claim (and a primary new finding compared to a prior study from this group, Anderson et al. 2019, Cell Reports), is that the identified remodeling clusters are not only CSF1R independent, but more specifically that expression of remodeling genes in response to neuronal apoptosis confers this survival advantage.

Two major concerns currently prevent this claim from being supported by the data shown:

1. Lack of biological replicates for single cell data: unless I misinterpreted the methods, there appears to be a single biological replicate for each single cell experimental condition. The authors rely on their sing cell data to measure the abundance of various microglial populations in their models. In the absence of replicates, it is not possible to confidently measure population abundance within and between samples, because there is no way to make an estimate of the error present in cluster sizes. By analogy, a single biological replicate of flow cytometry analysis could not be used to reliably quantify the size of populations. This reviewer does not want to force the authors to double or triple their single cell seq efforts, but this technical limitation really prevents reliable comparison of the abundance of cells in a given cluster within and across conditions, and these comparisons form the central basis for their claims. If not by more biological replicates of single cell data, a potential way to address this is to validate and better quantify the finding of shifted macrophage states using another method to show that remodeling cells go away in BAX ko, and are enriched with PLX treatment, and a possibility would be to use tissue staining or in situ to do this. Or maybe facs with some remodelign cluster markers?

2. Relationship between CSF1R independence and remodeling states: the authors conclude that remodeling states driven by neuronal apoptosis confer CSF1R independence. This conclusion is currently based on inferences from transcriptomic data about the functions of remodeling and PLX resistant microglia (eg that because they have a similar transcriptional profile, they are both engaged in remodeling), which is plausible but there are other equally plausible possibilities, such as that PLX directly induces a state change in microglia that does not involve neuronal apoptosis or remodeling. If the authors could directly show that surviving microglia are indeed engaged in some kind of a remodeling function, this would go a long way to support the claims. If not, careful adjustment of the claims and text would ensure that the claims are truly supported by the data shown, including a discussion of alternative explanations, and the limitations of transcriptomic inference to truly confirm that the "remodeling microglia" and the "plx resistant microglia" are a similar population of cells. This is particularly important because the major cluster in the plx3397 treated sample (cluster 3) is not present in any other condition.

*Reviewer #3 (Recommendations for the authors):*

The strength of the manuscript lies in its thorough and comprehensive characterization of multiple microglia states in the developing postnatal mouse retina. By the mean of scRNAseq analysis, the authors addressed the question of which environmental cues drive such heterogeneity.

This work allowed the identification of 11 microglia states that coexist in wild type postnatal retina, ranging from a so-called homeostatic state to some remodeling states, with for instance a disease-associated signature. The proportion of the different microglia states is context-dependent and the authors demonstrated that the spectrum of homeostatic to more remodeling clusters is driven predominately by neuronal apoptosis (resulting from waves of neuronal cell death during the postnatal period in the retina). On the other hand, recognition receptors, Mer or Axl, required for clearance of apoptotic cells and survival in the absence of CSF1R signaling, respectively, do not mediate changes in microglia remodeling gene expression.

The manuscript is extremely clear and the data are thoroughly described. They perfectly support the main conclusions. Although the finding that developmental apoptosis impact retinal microglia gene signature is not novel (a manuscript from the authors and based on bulk RNAseq analysis was published in Cell Reports in 2019), the comprehensive scRNA-seq clustering analysis is an important addition, providing a valuable resource for the field.

Some questions listed below should however be addressed for clarifications:

1. The authors nicely identified 11 clusters of microglia in the developing postnatal retina. This raises the question of the spatial distribution of these microglia within the retina and the dynamics during development. Although this goes beyond the scope of the manuscript, the authors show in Supp Figure 5 by in situ hybridization that Ccl3+ microglia are dispersed in the different retinal layers, suggesting a regular mosaic distribution of cluster 3. This question of spatial distribution of microglia clusters could be raised in the Discussion section.

2. On the right of cluster 1 in UMAP plots, there is a non-annotated blue cluster. What does it correspond to?

3. The authors found that density was reduced by nearly half in Bax KO. What is their interpretation of this data? Does this result from less microglia proliferation?

4. Statistical analysis would be useful in Figure 4D, Figure 3B and Figure 4B. Also, indicating ns on the graph for the 3 KO in Figure 6C would be helpful.

5. The authors conclude from their RNAseq analysis that there is no change in genes associated with survival pathways in Axl KO microglia compared to WT. However, they found that Spp1 is downregulated. Given that Osteopontin signaling is associated with survival in different cell types, why the authors do not consider it as a candidate?

6. From their RNAseq analysis, the authors conclude that Mer and Axl are not required to drive microglia remodeling gene expression. However, can the authors really exclude the possibility of post-transcriptional regulation?

7. Line 367, the authors conclude from their data that there are microglial states that are not regulated by neuronal death. Is it really the only interpretation of the data? Could it be a matter of threshold? If there were more apoptotic cells, would these microglial states still be not responsive?

8. The authors mention in the Discussion section that one candidate signaling downstream Axl is the PI3K-Akt-NfkB-Bcl2 pathway. It is not clear whether this is supported by the RNAseq analysis from the present manuscript.

---

## [Author Response]

Reviewer #1 (Recommendations for the authors):Reviewer concerns and suggestions for improvement:1. Additional methodological details about scRNAseq processing are needed.First and most importantly: In line 566 (Methods) the authors state that filtered cells from all 4 datasets were merged and then processed with the [Seurat] SCTransform pipeline. Were the 4 datasets combined using Seurat's integration/batch correction functionality, or were they simply merged? If the former, please provide in the Methods the details of the integration workflow. If the latter, the authors should explain the steps that were taken, if any, to mitigate batch effects. In the absence of such steps, there is a concern that batch effects, rather than true biological differences, could be the major driver of the differences in cluster composition between the WT/KO/vehicle/PLX samples.

No batch correction was performed on the single cell datasets. They were simply merged for dimensional reduction, and the methods have been updated to make this clear (lines 881). Batch effects were not obviously improved by batch correction with Seurat’s integration pipeline (see Author response image 1). Batch correction only served to separate Bax KO cells such that clearly similar mitotic cell populations no longer overlapped. While we acknowledge that there are differences between batches, we felt that the uncorrected data are more representative of the experiment.

**Author response image 1. sa2fig1:** 

2. Second, the authors do not state which count data from the Seurat data object were used to make figures and perform differential expression analysis. The standard in the field is to use log-normalized read counts. But since the SCTransform pipeline was used, the object will also contain "corrected" counts obtained via modeling. The methods should state which were used. If the latter were used the authors should justify this. It is particularly important to consider the appropriate methods for identifying differentially expressed genes (DEGs) because cluster 0, the major remodeling cluster, only had 4 upregulated DEGs that distinguished it from other clusters (Table S1). Two of these, Apoe and Ctsd, are important to the authors argument that cluster 0 cells have a remodeling phenotype. Further, some studies use lack of upregulated DEGs as an argument for "overclustering" i.e. that the clustering algorithm has split the data in a non-meaningful way. So a finding of zero DEGs would be significant for interpretation. For these two reasons it is critical to establish that the cluster 0 markers were identified in a sound manner.

Natural-log normalized counts were used for differential expression analysis and the methods have been updated to make this clear (lines 893-894). We have re-examined the DEGs defining the clusters, and found that the submitted Supplementary Table 1 was incomplete. We very much appreciate the Reviewer helping us to identify this error. Supplementary Table 1 (now called Supplementary File 1) has been updated to include additional upregulated and downregulated genes. It now lists 22 upregulated genes and 85 downregulated genes in Cluster 0.

3. Because Cluster 0 lacks definitive markers (Table S1), the reader is left wondering why exactly the clustering algorithm defined it as distinct from other clusters. The clear implication of the text is that cluster 0 is mainly defined by downregulation of homeostatic (cluster 1) markers. Four qualitative examples of this are given in Figure 2b. This point should be supported by a more systematic analysis. There are a number of ways this could be done, e.g. by showing DEGs downregulated in cluster 0 (compared to cluster 1 or to the entire dataset), or by plotting module scores for a larger set of homeostatic genes. Such an analysis is important to establish what genetic differences distinguish the cluster 0 cells, and whether they are biologically plausible or instead might indicate an overclustering artifact.

We thank the reviewer for drawing attention to Cluster 0. Our updated DEG list (Supplementary File 1) now includes 22 upregulated genes including remodeling markers (Apoe, Lyz2) and 85 downregulated genes including homeostatic marker Tmem119 and chemokines such as Ccl4 and Cxcl2. We are confident that this represents a distinct and relevant cluster, and not an overclustering artifact. We also added differential expression analysis between homeostatic (1 and 2) and remodeling clusters (0,3,4,5,6) to provide a list of differentially expressed genes (Supplementary File 2).

4. In the section on bulk RNAseq of Mer and Axl mutants, it is stated that there was no effect on lysosomal or lipid metabolism genes or DAM/ATM/PAM genes (lines 310-12 and 318-19). This is not clearly demonstrated by the data in Figure 7. It is great to see the heatmap listing all differentially expressed genes, but the authors should clarify that this list is consistent with the statements in the main text. (The reader cannot be expected to know what all of these genes do!) This could be done either by a gene ontology analysis or by explicitly providing a list of lysosomal and lipid metabolism genes that was used to compare with the differentially expressed gene list from Figure 7D. As for the DAM/ATM/PAM genes -- some are highlighted in the figures and text so this category is better supported. But I still think it would improve clarity and methodological transparency if the authors would provide a list of the genes they defined as DAM/ATM/PAM for this and other analyses used throughout the study.

We agree that comparison to a defined list would improve transparency and methodological rigor. We have addressed this by adding citations for GO lists of lysosomal and lipid metabolism genes and adding detail about DAM/ATM/PAM genes to the results (lines 170, 183-186, 188-191, 346-351, 354-356, 359-361). Further, we added a new section in the methods clearly defining these lists (lines 966-976). We used these lists to compare to bulk seq data in Figure 7 and scRNAseq in Figure 2.

5. Other scRNAseq studies using enzymatic dissociation have found clusters of ex-vivo activated microglia. See Marsh et al. 2020 biorxiv (cited by the authors). The fact that cells expressing this distinctive profile are missing is noteworthy and merits explanation. Were these cells detected and filtered out? Were transcriptional inhibitors used?

We agree that this is an important point. We did not filter out activated microglia or use transcriptional inhibitors during the dissociation. We cannot rule out ex vivo transcriptional changes, and do see expression of ex vivo activation (exAM) genes in some of our clusters. However, we found that ex vivo activation did not drive the formation of a distinct cluster as most of the expressed exAM genes were upregulated in multiple clusters. In addition, some of these genes were differentially expressed between samples. This may argue that these genes could have biological relevance. In support of this, histone-related “exAM” genes, for example, were predominately expressed in cycling microglia. When we compared our clusters to 27 exAM markers the authors used for scoring (Table 4 in Marsh et al. 2020), clusters 2 and 4 expressed the most exAM genes with 15/27 and 10/27, respectively. However, some of these genes have been shown to be expressed in vivo. We confirmed that two of these (Ccl3 and Ccl4) were expressed by in situ hybridization (Supplemental Figure 5 and not shown). Because the original study was in adult, the extent to which ex vivo transcription impacts developmental datasets or whether the same signature would emerge, are currently unknown. However, we cannot rule out ex vivo transcriptional changes and have added a sentence to the discussion (lines 427-432).

6. The conceptual explanation for the Bax in silico genotyping strategy needs to be expanded, either in the supplementary figure legend or Methods. I appreciated the supplementary figure, but I still don't really understand what the overall strategy was or what the plots in Figure S3C-F are showing. Why was it expected that BaxWT and KO would have distinct SNPs? How were the SNPs identified (at a conceptual level; the explanation of software tools is reasonably clear)? Why are Ftl1 and Ftl-ps1 part of this analysis?

We agree that the in-silico genotyping strategy was not well explained. We have made changes to Supplementary Figures 2 and 3 in an effort to make the strategy and results clearer. We have expanded our explanation of this process in the methods (lines 897-925), and added a brief note on why Bax-linked genes, Ftl1 and Ftl1-ps1, were considered in these analyses (lines 883-888), but we have removed the plots.

7. The plots in Figure 2B-E (UMAP Feature Plots) give a qualitative sense of how these genes change across clusters. The authors' argument would be strengthened by pairing this with a more quantitative view of how gene expression differs between homeostatic and remodeling groups. E.g. by using dot plots, violin plots, or explicit tests for up/downregulated genes comparing of homeostatic vs. remodeling groups.

We agree with the reviewer that Figure 2 would be strengthened by a more quantitative view. Therefore, we added violin plots of select clusters highlighting genes of interest (Figure 2C’,D’,E’, F’). We also added differential expression analysis between homeostatic (1 and 2) and remodeling clusters (0,3,4,5,6,10) Supplementary File 2.

8. The analysis of apoptotic clearance in Mer and Axl mutants is compelling. However only one time point, near the end of the death period, was checked (P5). Do apoptotic bodies persist in these mutants or can other mechanisms compensate for the loss of Mer/Axl?

We thank the reviewer for the interesting question. We think other mechanisms likely compensate in the absence of Mertk or both Mertk/Axl since microglia express a broad array of phagocytic receptors and other cell types such as astrocytes can phagocytose, particularly in the context of deficient microglia (Puñal et al. 2019). However, experiments looking at later timepoints would be difficult to interpret due to cell death of RGCs until P10, albeit at low levels, as well as the continued breakdown of apoptotic cells into smaller and smaller bodies. We added a sentence clarifying this point in the discussion (lines 442-445).

Reviewer #2 (Recommendations for the authors):In the article titled "Neuronal apoptosis drives remodeling states of microglia and shifts in survival pathway dependence" the authors describe the heterogeneity of the microglial population in the early postnatal retina. In WT mice, the authors find both "homeostatic" microglia (expressing Tmem119 and P2RY12) and several clusters of "remodeling" microglia (expressing lysosomal and phagocytic components). In a Bax KO mouse lacking the significant apoptosis that usually occurs in the early postnatal retina, they find a decrease in the remodeling clusters of microglia, concluding that the lack of apoptosis in this mouse prevents the shift in microglial phenotype from homeostatic to remodeling. Several of these "remodeling" clusters in WT mice are relatively enriched after short term PLX3397 treatment, a CSF1R inhibitor commonly used to deplete microglia in mouse.This paper provides interesting new phenotyping of retinal microglia and how they change with Bax KO and PLX treatment, but one central conclusion is currently not fully supported by the data provided. The central claim (and a primary new finding compared to a prior study from this group, Anderson et al. 2019, Cell Reports), is that the identified remodeling clusters are not only CSF1R independent, but more specifically that expression of remodeling genes in response to neuronal apoptosis confers this survival advantage.Two major concerns currently prevent this claim from being supported by the data shown:1. Lack of biological replicates for single cell data: unless I misinterpreted the methods, there appears to be a single biological replicate for each single cell experimental condition. The authors rely on their sing cell data to measure the abundance of various microglial populations in their models. In the absence of replicates, it is not possible to confidently measure population abundance within and between samples, because there is no way to make an estimate of the error present in cluster sizes. By analogy, a single biological replicate of flow cytometry analysis could not be used to reliably quantify the size of populations. This reviewer does not want to force the authors to double or triple their single cell seq efforts, but this technical limitation really prevents reliable comparison of the abundance of cells in a given cluster within and across conditions, and these comparisons form the central basis for their claims. If not by more biological replicates of single cell data, a potential way to address this is to validate and better quantify the finding of shifted macrophage states using another method to show that remodeling cells go away in BAX ko, and are enriched with PLX treatment, and a possibility would be to use tissue staining or in situ to do this. Or maybe facs with some remodelling cluster markers?

We sequenced a single technical replicate for each condition, but each consisted of several biological specimens (11-13 animals each) from multiple litters, totaling 49 animals. Pooling this large number of retinas was necessary to obtain enough cells for analysis, and consequently, we have many biological samples represented. In addition, the proportions of microglia expressing specific markers in our scRNAseq match our published quantification by in situ hybridization (Cell Reports 2019), giving us confidence that the proportions determined here are representative. For example, we previously found that in P7 WT retinas, 100% of microglia express Apoe and Lyz2 at varying levels, 65% Tmem119, and 25% Spp1 and these numbers are consistent with proportions identified by our scRNAseq. The same is true for both the Bax KO and PLX treated retinas. We thank the reviewer for pointing this out and have emphasized the number of animals/retinas used in the results (line 110-112), figure legend and methods (lines 856-860) and importantly, that these proportions match our previous quantification (results line 158-160).

2. Relationship between CSF1R independence and remodeling states: the authors conclude that remodeling states driven by neuronal apoptosis confer CSF1R independence. This conclusion is currently based on inferences from transcriptomic data about the functions of remodeling and PLX resistant microglia (eg that because they have a similar transcriptional profile, they are both engaged in remodeling), which is plausible but there are other equally plausible possibilities, such as that PLX directly induces a state change in microglia that does not involve neuronal apoptosis or remodeling. If the authors could directly show that surviving microglia are indeed engaged in some kind of a remodeling function, this would go a long way to support the claims. If not, careful adjustment of the claims and text would ensure that the claims are truly supported by the data shown, including a discussion of alternative explanations, and the limitations of transcriptomic inference to truly confirm that the "remodeling microglia" and the "plx resistant microglia" are a similar population of cells. This is particularly important because the major cluster in the plx3397 treated sample (cluster 3) is not present in any other condition.

We agree that function cannot be determined by transcriptomic data alone. The conclusion that remodeling states driven by neuronal apoptosis confer CSF1R independence is bolstered by our previous publication where we found that in the Bax KO animal, a larger proportion of microglia were dependent on CSF1R signaling (Cell Reports 2019) referenced in lines 482-484 of the discussion. We are currently exploring whether “resistant” populations are actively engaged in remodeling activities, which is being prepared for another publication. However, we agree that other possibilities exist for the overlap in PLX-resistant and remodeling clusters and have added alternative explanations in the discussion (line 492-495) and have qualified this conclusion in results (line 270).

Reviewer #3 (Recommendations for the authors):Some questions listed below should however be addressed for clarifications:1. The authors nicely identified 11 clusters of microglia in the developing postnatal retina. This raises the question of the spatial distribution of these microglia within the retina and the dynamics during development. Although this goes beyond the scope of the manuscript, the authors show in Supp Figure 5 by in situ hybridization that Ccl3+ microglia are dispersed in the different retinal layers, suggesting a regular mosaic distribution of cluster 3. This question of spatial distribution of microglia clusters could be raised in the Discussion section.

We agree that the spatial distribution of these various populations is of interest since it may help inform function. This is a focus of ongoing investigation. We previously examined the distribution of Apoe, Lyz2, Clec7a, Spp1, Igf1, Tmem119, and Itgax (Cell Reports 2019). While we found that these genes were dynamically regulated over development, a detailed localization analysis at one time point was not done. We agree this would be important and informative. We have added a sentence to the discussion (line 398-400).

2. On the right of cluster 1 in UMAP plots, there is a non-annotated blue cluster. What does it correspond to?

Those cells are members of cluster 6. We added a note in the figure legend of Figure 1.

3. The authors found that density was reduced by nearly half in Bax KO. What is their interpretation of this data? Does this result from less microglia proliferation?

The observation that microglial density is reduced in the Bax KO is an intriguing one. We currently do not know whether it is a result of less proliferation, but we do not see a change in the proportion of cycling microglia by scRNAseq in KO microglia compared to WT at P7 (Figure 3A,B). We added a sentence to the discussion (lines 403-407).

4. Statistical analysis would be useful in Figure 4D, Figure 3B and Figure 4B. Also, indicating ns on the graph for the 3 KO in Figure 6C would be helpful.

We have added statistical information in the figure and legend for Csf1r expression across clusters as well as a Chi-square test looking at the change in distribution between samples for Figures 3B and 4B. In addition, we have added “ns” for comparisons that are not significant to Figure 6C.

5. The authors conclude from their RNAseq analysis that there is no change in genes associated with survival pathways in Axl KO microglia compared to WT. However, they found that Spp1 is downregulated. Given that Osteopontin signaling is associated with survival in different cell types, why the authors do not consider it as a candidate?

We thank the reviewer for this excellent point and for identifying this oversight. We have added several sentences to the discussion highlighting this possibility (lines 519-524).

6. From their RNAseq analysis, the authors conclude that Mer and Axl are not required to drive microglia remodeling gene expression. However, can the authors really exclude the possibility of post-transcriptional regulation?

We focus here on a set of remodeling genes that we know can be transcriptionally regulated in our context as well as others. However, we agree that post-transcriptional regulation is likely involved in driving many microglial responses as well, and cannot rule out the involvement of Mer and Axl. We have added a sentence to the discussion (line 462-463).

7. Line 367, the authors conclude from their data that there are microglial states that are not regulated by neuronal death. Is it really the only interpretation of the data? Could it be a matter of threshold? If there were more apoptotic cells, would these microglial states still be not responsive?

Good point. We changed the text to reflect that an alternative possibility is that these clusters represent cells that have not had the opportunity to interact with dying cells (lines 420-421).

8. The authors mention in the Discussion section that one candidate signaling downstream Axl is the PI3K-Akt-NfkB-Bcl2 pathway. It is not clear whether this is supported by the RNAseq analysis from the present manuscript.

This pathway is primarily regulated post-transcriptionally and thus, we don’t expect to see changes at the gene expression level.